# Self-Calibrating BCIs: Ranking and Recovery of Mental Targets Without Labels

**Jonathan Grizou**∗
GrizAI
University of Glasgow
jonathan.grizou@grizai.com

**Carlos de la Torre-Ortiz**∗
University of Helsinki
carlos.delatorreortiz@helsinki.fi

**Tuukka Ruotsalo**
LUT University
University of Copenhagen
tuukka.ruotsalo@lut.fi

## Abstract

We consider the problem of recovering a mental target (e.g., an image of a face) that a participant has in mind from paired EEG (i.e., brain responses) and image (i.e., perceived faces) data collected during interactive sessions without access to labeled information. The problem has been previously explored with labeled data but not via self-calibration, where labeled data is unavailable. Here, we present the first framework and an algorithm, CURSOR, that learns to recover unknown mental targets without access to labeled data or pre-trained decoders. Our experiments on naturalistic images of faces demonstrate that CURSOR can (1) predict image similarity scores that correlate with human perceptual judgments without any label information, (2) use these scores to rank stimuli against an unknown mental target, and (3) generate new stimuli indistinguishable from the unknown mental target (validated via a user study, $N = 53$). We release the brain response data set ($N = 29$), associated face images used as stimuli data, and a codebase to initiate further research on this novel task.

**Website** — https://jgrizou.github.io/neurips-self-calibrating-bci/
**Code and Data** — https://github.com/jgrizou/neurips-self-calibrating-bci/

## 1 Introduction

We tackle the problem of recovering a mental target (e.g., an image of a face) that a participant holds in mind, using only unlabeled EEG responses (i.e., brain activity) recorded while the participant views a sequence of unlabeled images (i.e., perceived faces). These images, along with the mental target, are represented as high-dimensional vectors in the latent embedding space of a generative model. Thus, unlike traditional brain-computer interfaces (BCIs) that classify brain responses into a few discrete classes (e.g., yes/no or up/down/left/right), our task requires mapping EEG responses to distances in this high-dimensional space in order to recover the mental target iteratively. The challenge lies in probing these *continuous* high-dimensional spaces (image and EEG spaces) without discretizing them into labeled classes and only relying on paired data of image and brain representations.

Previous BCI methods operating in continuous domains — such as decoding a distance to a mental target — required an explicit calibration phase relying on labeled data (i.e., known targets) to train a

---

∗equal contribution

39th Conference on Neural Information Processing Systems (NeurIPS 2025).

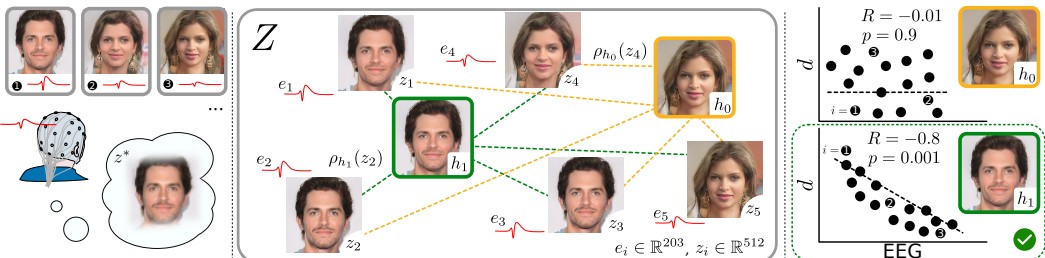

Figure 1: (**Left**) Participants are instructed to hold a mental target $x^*$, represented as $z^*$, unknown to our system, while a sequence of $i = \{1, \ldots, N\}$ images are shown to them and their EEG responses are being recorded. CURSOR's goal is to recover $z^*$ with this information alone and fully unsupervised. (**Middle**) CURSOR generates hypothetical target images, e.g. $h_1$, and builds a dedicated estimation problem to predict the distance $d_i$ between $h_1$ and $z_i$ from the corresponding EEG response $e_i$, for all $i$. (**Right**) The performance of an estimator on this task is the score $S(h_1)$ attached to $h_1$. The hypothesis with the highest score $\hat{h} = \arg\max_{h \in \mathcal{H}} S(h)$ is assumed to be the unknown mental target. Images are generated from $z_i \in \mathbb{R}^{512}$ embeddings with EEG responses $e_i \in \mathbb{R}^{203}$. We collected $N = 9234$ $\{z_i, e_i\}$ pairs for this study from 29 participants.

decoder [71]. In contrast, this paper introduces a method that requires no prior supervision or labeled examples, which we term fully *self-calibrating*. Self-Calibrating BCIs (SC-BCIs), in general, are those that can infer the decoding relationship without an experimenter-provided labeled training set [31, 23]. Prior SC-BCI research, however, has primarily focused on tasks involving a discrete stimulus set and categorical feedback (e.g., P300 spellers, where the target is one option from a small known set, and feedback is match/non-match) [38, 25, 32, 31, 70]. The challenge of self-calibrating in continuous domains is significantly greater, requiring a search over a theoretically infinite latent stimulus space using a continuously-valued feedback decoded from brain signals (e.g., a latent-space distance or other regressed value, in contrast to a discrete match/non-match classification). To our knowledge, this paper presents the first SC-BCI framework designed for continuous domains. Our method, CURSOR, achieves label-free self-calibration and iteratively estimates the unknown mental target, thus implicitly recovering the necessary information (i.e., labels) for subsequent decoder training. We demonstrate the effectiveness of CURSOR in a mental target recovery task.

To enable our study, we collected a large dataset ($N = 9234$) of EEG responses ($\mathbb{R}^{203}$) recorded as participants ($N = 29$) viewed face images generated from a latent embedding space ($\mathbb{R}^{512}$) while holding a mental target image in mind. Collecting novel data was necessary because existing datasets [19, 26] focus on categorization tasks with participants passively observing stimuli. Compared to existing datasets, ours is, to our knowledge, the only existing dataset suitable for this task because (1) stimuli are represented within a continuous space, (2) participants actively engage with a target in mind, (3) quantitative distance metrics between stimuli can be defined, and (4) ground-truth data for human distance judgment between two stimuli are obtained through a user study.

Equipped with a novel suitable dataset, we propose CURSOR (Consistency-based Unsupervised Regression for Similarity-based Optimization and Ranking), the first self-calibrating BCI algorithm for continuous domains. As illustrated in Figure 1, CURSOR first generates a hypothetical mental target as an embedding vector representing a face a user might have in mind. Given this hypothesis, CURSOR constructs a hypothesis-specific dataset that transforms observed faces into distances to this hypothetic target (Alg. 1, line 2). CURSOR then trains an estimator to decode EEG responses into these hypothetical distances. The performance of this estimator, relative to that of one trained on a deliberately misaligned dataset, defines the CURSOR score for the hypothesis under consideration (Alg. 1, line 8). The CURSOR score should peak when evaluated at the (unknown) true mental target, because the EEG responses were recorded in that context and, therefore, should provide the strongest basis for predicting the corresponding distances.

CURSOR does not require access to any ground-truth supervision labels and thus solves the SC-BCI challenge. Our experiments confirm that CURSOR can identify the mental target a user has in mind from unlabeled EEG responses without requiring calibration, pre-trained decoders, heuristics, or labels. To our knowledge, this is the first work extending SC-BCIs to continuous domains.

Our contributions are as follows:

1. We introduce the first SC-BCI framework and associated dataset for continuous domains.

2. We propose CURSOR, the first SC-BCI algorithm for continuous domains, supported by empirical validation across ranking and optimization tasks, allowing generating the mental image target and recovering ground truth labels.

3. We empirically analyze CURSOR's performance under various conditions, including estimator type, data representation, and sensitivity to the amount of data.

## 2   Related Work

**BCI-driven visual decoding tasks.**   We build on the established distinction between passive decoding (reconstructing viewed stimuli) and the active, iterative search of a mental image target [36].

In **passive EEG-guided generative decoding**, EEG is recorded while a participant views *unseen* images, and each signal is projected into a semantic latent space (e.g., CLIP or diffusion) from which a representative image is synthesized [4, 27]. The training uses time-aligned EEG–image pairs *without* category labels and can scale well with self-supervised objectives. Nevertheless, these methods typically generate an image from the target class rather than the *specific* stimulus instance (e.g., *some* face, not the exact face identity). Some works introduced fixed, block-structured stimuli resulting in temporal confounds that have been shown to inflate accuracy [2, 45, 1, 7]. Our approach avoids these confounds and does not belong to the passive decoding family.

For **iterative mental target search**, the system updates its estimate of the mental target based on preceding EEG responses, progressively steering the stimulus sequence toward a single mental target (e.g., a specific face image). Previous work relied on labeled target trials to learn the EEG-driven update rule and adapt the target estimate with heuristics based on binary labels [36] or distance estimates [71] decoded from EEG. Our method preserves the iterative search behavior of the latter while *eliminating the label requirement*, recovering a concrete target within a continuous high-dimensional generative manifold without pre-training data *or* ground-truth labels at any stage.

Although our work adheres to the iterative search framework, we do not report results from a fully closed-loop task [32, 60]. Instead, our evaluation is conducted offline (iterative but non-interactive) using EEG recorded from a pre-generated stimulus sequence. All iterative analyses are performed post-hoc. Moreover, we do not employ active sampling; EEG–image pairs are drawn randomly without replacement from the offline dataset.

**SC-BCIs in classification contexts.**   Self-calibrated BCIs (SC-BCIs) [31, 23] have been primarily developed for classification tasks. For instance, [38] relied on Expectation-Maximization (EM), assuming signals can be linearly projected onto two one-dimensional Gaussian distributions (yes/no responses) and aiming to maximize the distance between these distributions. Later designs used classification accuracy on datasets attached to each possible target as a scoring metric [25, 32], which [67, 70] approximated using the distances between the mean of two classes. Label proportion (LP), initially proposed by [59], was applied to BCIs by [30]. LP assumes a known, unbalanced ratio of labels in stimuli-response pairs that can be exploited to label clusters identified from unsupervised clustering methods. Further work has extended LP to reinforcement learning [82] from unlabeled reward to break symmetry after applying contrastive learning, and [74] explored the combination of EM and LP methods.

Self-calibration was also explored in Human-Robot Interaction [24], Human-Computer Interfaces [61, 23, 81], Security and Privacy [47] and Language Games [76, 5], always in a classification context.

## 3   Self-calibration Problem Formulation

We consider interaction sessions as pairs of stimuli and responses, where stimuli are images of faces $x \in X$, represented as embedding vectors $z \in Z$ from a known generative model $G : Z \mapsto X$. We rely on the same model to collect paired (x, z) for simplicity, but other methods could be used to generate and embed stimuli, respectively, which could be part of future research. User responses are EEG signals $e \in E$, recorded post-stimuli. An interaction session as a sequence of $N$ stimuli-response

pairs $\Psi_N = \{(z_i, e_i)\}_{i=1}^N$. During interactions, participants guide the system by mentally focusing on a target $x^*$, represented as $z^*$ and unknown to the algorithm, while they observe a sequence of stimuli $x_i$, represented as $z_i$. The aim is to estimate $\hat{z}$ that approximates $z^*$ using $\Psi_N$.

**Similarity equivalence.** Participants observe only $x_i$ without explicit knowledge of $Z$, but we know EEG responses $e_i$ encode perceptual similarities in $X$, correlating with distances $d_i$ in $Z$ [72]. In other words, EEG responses $e_i$ do not encode $z^*$ directly but "how far" $z$ is from $z^*$. This implies the existence of a function $f_{\theta_{z^*}} : E \mapsto \mathbb{R}^+$ mapping EEGs $e_i$ to distances $d_i$ in $Z$, although it is not directly accessible. Similarly, we assume the existence of a similarity function $\rho_{z^*} : Z \to \mathbb{R}^+$ that returns a distance $d_i$ between a mental target $z^*$ and an observed stimulus $z_i$. Hence:

$$f_{\theta_{z^*}}(e_i) = d_i = \rho_{z^*}(z_i) \tag{1}$$

with $z^*$, $\theta_{z^*}$, $d_i$, $\rho_{z^*}$ unknown but access to $(e_i, z_i)$ pairs.

**Error estimation.** According to Equation 1, the task reduces to identifying $\hat{z}$ that minimizes the error between our two estimates $f_{\theta_{\hat{z}}}(e_i)$ and $\rho_{\hat{z}}(z_i)$, based on observed $(z_i, e_i)$ pairs in $\Psi_N$. This can be measured on the entire dataset by defining an error-based score such as:

$$\hat{z} = \arg\min_z \underset{i=1..N}{ERROR}(f_{\theta_z}(e_i), \rho_z(z_i)) \tag{2}$$

where $ERROR()$ is a metric function defined to best suit the problem.

The SC-BCI challenge lies in the reciprocal dependency between $\theta_z$ and $\rho_z$. Both are apriori unknown and need to be learned. But to learn $\theta_z$, we need to access $\{e_i, d_i\}$ pairs, which can only be acquired by knowing $\rho_z$ to reconstruct $d_i$ based on $z_i$; and vice versa, to learn $\rho_z$, we need $\theta_z$.

# 4 CURSOR Algorithm

CURSOR, our proposed algorithm, solves this SC-BCI challenge by (1) defining a similarity function $\rho_z$ for any $z$, which allows us to break the dependency deadlock in Equation 2; (2) defining an $ERROR$ function that controls for the variability between datasets generated via $\rho_z$; and (3) searching the hypothesis space of mental targets $Z$ guided by the $ERROR$ scores.

In the following, we use $h \in \mathcal{Z}$ as a hypothetic mental target to be evaluated in Equation 2. We use $h$ to differentiate hypotheses from observations $z_i$ clearly, ground truth mental target $z^*$, and best estimate $\hat{z}$, which all exist in $Z$.

**(1) Defining a similarity function.** We define $\rho_h(z_i) = \|h - z_i\|_2$, which we assume to be valid for any hypothesis target $h$ and any observed stimuli $z_i$. This choice reflects the assumption that the perceptual similarity between an observed image $z_i$ and a target image $\hat{z}$, as encoded in their EEG responses, correlates linearly with Euclidean distances in the structured latent space $Z$, in agreement with the findings of [72]. Knowing $\rho_h$ allows the estimation of Equation 2 by breaking the reciprocal dependency between $\theta_h$ and $\rho_h$ for a given $h$. We use $\rho_h(z_i)$ to build a first estimate of $d_i$, which in turn can be used to estimate $\theta_h$ via $(e_i, d_i)$ pairs.

**(2) Defining a $ERROR$ function.** Given $\rho_h(z_i)$, we construct a dataset $\Gamma_N^h = \{(e_i, d_i^h)\}_{i=1}^N$ associated to $h$ where $d_i^h = \rho_h(z_i)$ (Alg. 1, line 2). With this dataset, we can estimate $\theta_h$ by training an estimator on $\Gamma_N^h$. Equipped with both $f_{\theta_h}$ and $\rho_h$, we can rely on well established error metrics for Equation 2, such as $\text{RMSE}(f_{\theta_h}(e), \rho_h(z))$, where lower values indicate stronger predictability.

However, comparing RMSE scores directly is unsuitable because the distribution of $d$ values in $\Gamma_N^h$ will vary between hypothesis $h$, even post-standardization. For example, a hypothesis $h_f$ "far away" from all observed $z_i$ would lead to a narrow distribution of $d_i^{h_f}$, which is highly predictable. Hence, an RMSE score based on $\Gamma_N^{h_f}$ could compare favorably even with respect to the ground truth dataset $\Gamma_N^{z^*}$.

To standardize errors across datasets, we compute a *relative gain* against a baseline score computed from misaligned stimuli-response pairs of the same dataset. In practice, a permutation $\sigma$ produces a

shuffled dataset $\Gamma_{N,\sigma}^h$ (Alg. 1, line 5). The final CURSOR score $S(h)$ is defined by the ratio of the respective RMSE scores (Alg. 1, line 8) as:

$$S(h) = \frac{\text{RMSE}(f_{\theta_{h,\sigma}}(e_\sigma), \rho_h(z))}{\text{RMSE}(f_{\theta_h}(e), \rho_h(z))} = \frac{\|f_{\theta_{h,\sigma}}(e_\sigma) - \rho_h(z)\|_2}{\|f_{\theta_h}(e) - \rho_h(z)\|_2} \tag{3}$$

with $S$ a relative error-ratio function which is to be maximized so $\hat{h} = \arg\max_h S(h)$.

Although we instantiate Eq. 3 with RMSE, other standard choices, such as negative log-likelihood, are in principle equally viable. In our setup, estimators are primarily point predictors and squared-loss training aligns naturally with RMSE, which we adopt for simplicity and consistency. Crucially, our *relative gain* $S(h)$ is designed to neutralize distributional differences across $h$.

---

**Algorithm 1** CURSOR: Scoring Function

---

**Require:** Stimuli–response dataset $\Psi_N = \{(z_i, e_i)\}_{i=1}^N$
**Require:** Target hypothesis $h \in \mathcal{Z}$
**Require:** Similarity function $d_i^h = \rho_h(z_i) = \|h - z_i\|_2$
**Require:** Estimator $f_\theta : E \to \mathbb{R}^+$
**Require:** Error function $\text{RMSE}(\hat{y}, y^*) = \frac{1}{\sqrt{N}}\|\hat{y} - y^*\|_2$

1: **function** $S(h)$
2:      Construct hypothesis dataset $\Gamma_N^h = \{(e_i, d_i^h)\}_{i=1}^N$
3:      Train estimator $f_{\theta_h}$ on $\Gamma_N^h$
4:      Compute $\text{RMSE}_h = \text{RMSE}(f_{\theta_h}(e), d^h)$
5:      Generate shuffled dataset $\Gamma_{N,\sigma}^h = \{(e_{\sigma(i)}, d_i^h)\}_{i=1}^N$
6:      Train shuffled estimator $f_{\theta_{h,\sigma}}$ on $\Gamma_{N,\sigma}^h$
7:      Compute shuffled $\text{RMSE}_{h,\sigma} = \text{RMSE}(f_{\theta_{h,\sigma}}(e_\sigma), d^h)$
8:      **return** $\text{RMSE}_{h,\sigma}/\text{RMSE}_h$

---

**(3) Searching the hypothesis space.** CURSOR scores can be used in downstream tasks such as ranking and optimization. Indeed, hypotheses $h$ *do not* need to be among observed stimuli or within a fixed test set, allowing any stimulus representation in $Z$ to be scored, thus allowing $S$ to guide an optimization process throughout the latent space.

## 5 Neurophysiological Data Acquisition

To run our experiments, a dataset of artificial human face images and associated EEG responses was collected from 31 participants, with 2 dropped due to data corruption.

**Generating stimuli for the experiment.** A pretrained GAN [20] on the CelebA-HQ dataset [37][2] was used to generate stimuli images. The GAN maps its latent space $Z$ in $\mathbb{R}^{512}$ to the image space $X$ as $G : Z \mapsto X$. We sampled the generator using 512-dimensional Gaussian noise to produce images and selected 17 target images without visual artifacts, represented by $z_t^*$ latent vectors. For each target, stimuli were generated by sampling along a diagonal trajectory in the 512-dimensional latent space, moving equally far in every coordinate direction by adding a small, monotonically increasing positive value to each component of the source image latent representation. This approach is an established best practice in face research to ensure small, smooth, and monotonic changes in facial appearance between stimuli [73, 62, 35, 22]. The sampling itself employed logarithmic spacing (denser near 0). This choice is motivated by its utility in BCI and psychometric paradigms [78, 80], as it efficiently focuses data acquisition near the target, where the resulting changes in brain activity are expected to be most sensitive to small stimulus variations [43, 13].

Although the standard logarithmic sampling strategy ensures data quality, we recognize it introduces an acquisition bias that limits direct expansion to online closed-loop settings. We stress-tested for this potential bias by: (i) evaluating the ranking task using uniformly sampled hypotheses, which are decoupled from the log-biased acquisition; and (ii) conducting ablation studies where near-target stimuli were systematically removed (Figure 4 and Appendix C.1).

**BCI task, EEG processing, and dataset.** Data consist of EEG responses elicited while subjects viewed images at varying similarity distances to a cued target in an RSVP paradigm. Before the RSVP stream, subjects were shown the target and instructed to make a mental note of it. Images then

---

[2]Pre-trained model and code: `https://github.com/tkarras/progressive_growing_of_gans`

appeared every 500 ms, aiming to evoke a P3-like response. EEG was recorded using a 32-channel EasyCap system (29 effective channels after removing corrupted ones) and signals were processed offline using MNE for Python [21]. Standard filtering, time-locking into "EEG epochs," and artifact rejection were applied. Temporal EEG data was windowed into 7 equidistant time windows within the 50–800 ms range after stimulus onset. Within each of the 7 windows, samples are averaged across time for each channel (after baseline correction), producing one scalar per (channel, window); stacking across 29 channels and 7 windows results in a $\mathbb{R}^{203}$ vector. Our final dataset consists of 9234 stimuli-response pairs, with each entry composed of a $z_i^*$ vector in $\mathbb{R}^{512}$, a stimuli $z_i$ vector in $\mathbb{R}^{512}$, and an EEG response vector $e_i$ in $\mathbb{R}^{203}$. *The complete details can be found in Appendix A.*

# 6    Experiments

We evaluated CURSOR's ability to rank and optimize facial stimuli against an unknown mental target based on unlabeled EEG responses. We describe our experimental framework, dataset preparation, and comparative studies, including ablations and control conditions. In addition, we validated our approach through human perception studies.

## 6.1    Datasets preparation for simulation runs

The original dataset comprise stimuli-response pairs (stimulus $z_i$, EEG response $e_i$) collected across multiple experimental sessions and different participants, meaning not all pairs were associated with the same target face. To create a sufficiently large, unified dataset for our simulation runs — where every pair is associated with a single target face — we generated equivalent datasets. Given previous work confirming that the EEG response primarily encodes the Euclidean distance $d(z_i, z^*)$ in the latent $\mathcal{Z}$-space [72], we could generate new stimuli $z_i'$ that preserved the original statistical properties for any new target $z_{new}^*$. Starting from a random 512-D initialization, we used a second-order numerical solver [50, 75] to minimize the distance-matching objective and recover $z_i'$ at distance $d_i$ around $z_{new}^*$, such that $d(z_i', z_{new}^*) \approx d(z_i, z_{original}^*)$. Specifically, we created 10 dataset variants for each of the 17 target faces utilized in our neurophysiological experiments, yielding 170 stimuli-response datasets, each containing N=9234 pairs.

For ranking tasks, we built separate hypothesis sets of $L = 60$ faces for each of the 17 target faces. Each hypothesis $h_l$ is generated at distance $d_l$ from the target, with $d_l$ sampled uniformly in $[0, 46.16]$. $46.16$ being our dataset's maximum $d$ value, ensuring our evaluation is within observed bounds.

## 6.2    Evaluation frameworks

Our study faces computational challenges due to high-dimensional data: stimuli (image embedding representation) in $\mathbb{R}^{512}$ and responses (flattened EEG representations) in $\mathbb{R}^{203}$ making search a difficult process without a large sampling budget. Evaluating $S$ is also computationally intensive, requiring cross-validation training of estimators on the 203D response space at each iteration. The need for multiple runs, baselines, and ablation studies compounds these challenges. We thus adopted a two-stage evaluation approach: ranking a predefined set of hypotheses and then optimizing in dimensionally-reduced spaces.

**Ranking a predefined set of hypotheses.** For ranking evaluations, we assume the system has access to a finite set of $L$ hypotheses $\mathcal{H} = \{h_1, \ldots, z^*, \ldots, h_L\}$, which allows us to compute score on a finite set of candidates. To simplify the interpretation of results, we included the mental target $z^*$ in $\mathcal{H}$ which guarantees the oracle baseline of the "similarity to top rank" metrics is 0 on Figure 3 right. This is by no means a requirement of our approach as we also demonstrate optimization results on dimensionally reduced spaces (see Figure 4). We rank the hypotheses directly by their CURSOR's scores, breaking ties randomly. The following metrics are then evaluated:

- **Score-distance correlation.** The Pearson's correlation coefficient $R(S(h), \rho_{z^*}(h))$ between CURSOR scores and ground-truth target distances.
- **Target rank.** The rank of the target stimuli as measured by $rank(z^*) = \sum_{h \in \mathcal{H}} \mathbb{I}(S(h) \geq S(z^*))$ where $\mathbb{I}(\cdot)$ is the indicator function.
- **Similarity at top rank.** The similarity of the best ranked hypothesis to the ground-truth target as $\rho_{z^*}(\hat{h})$.

**Optimizing in dimensionally-reduced spaces.** To demonstrate the feasibility of optimizing a target stimulus based on our scoring function $S$, we reduced the dimensions of $e$ for faster score calculation and of $z$ for faster convergence. We applied Principal Component Analysis (PCA) to reduce $e$ to $\mathbb{R}^{20}$ and $z$ to $\mathbb{R}^{10}$, with Appendix B detailing our rationale with computational evidence. We conducted optimization experiments in these reduced spaces. Covariance Matrix Adaptation Evolution Strategy [28], implemented via Optuna [3] using default parameters, was chosen for optimization due to the high-dimensional, gradient-free nature of our problem. Optimization bounds were set to $[-15, 15]$ per reduced dimension, corresponding to maximum distances in the 512D space that align with our ranking experiments (47.42 vs 46.16), allowing a fair comparison between ranking and optimization performance. We report the similarity of the optimized stimuli with the ground truth target $\rho_{z*}(\hat{h})$ projected back to the original $Z$ to allow a meaningful comparison with the ranking and human-experiment results.

## 6.3 Comparative and ablation studies

For a comprehensive evaluation, we evaluated the impact of different estimator models, included two control conditions, ran multiple data scarcity scenarios, and tested with two EEG signal representations. The computational environment is detailed in Appendix C.3.

**Estimators.** Our method requires training estimators $f_{\theta_h}$ on $\Gamma_N^h$ to compute the Self-Calibration score $S(h)$. We compared performance using Linear Regression (LR), Support Vector Regression (SVR), and Multi-Layer Perceptron (MLP) as implemented in the Scikit-learn library [56]. The relative model simplicity allows for rigorous evaluation within our computational constraints, and linear models are known to perform well in EEG tasks [55, 69, 9, 42, 8]. When training estimator parameters $\theta$, both EEG $e$ and distances $d$ are standardized by removing the mean and scaling to unit variance for each feature.

Each $S(h)$ estimate is the average of a 10-fold 90%/10% randomly partitioned cross-validation procedure. This cross-validation is essential for two reasons: (1) it ensures the estimators do not train and test on the same data, and (2) it provides the held-out data necessary to compute the relative scores that define $S(h)$. We specifically address potential overfitting by computing $S(h)$ as a ratio of errors. Furthermore, for each cross-validation fold, the aligned RMSE and the shuffled RMSE are computed on the identical held-out fold. Thus, if an estimator overfits or performs poorly, the aligned and shuffled errors should be similar, resulting in $S(h) \approx 1$, which is easily detected and can prevent premature decisions in online settings.

As detailed in Appendix C.1, we decided not to pre-tune estimators' meta-parameters using ground-truth information because our approach is meant to be fully unsupervised, and doing so would not represent a realistic SC-BCI scenario. Nonetheless, we assessed the performance of SVR and MLP with offline hyperparameter optimization, which we detail in Appendix E.1.

**Control conditions.** We benchmarked CURSOR using theoretically ineffective estimators for computing $S(h)$. Our control conditions include: 1) a dummy estimator (*Dummy*) predicting $d$ as the average of all distances in $\Gamma_N^h$ without considering $e$, and 2) a Linear Regressor that always Shuffle stimuli and response pairs prior to training (*S-LR*). Both baselines should perform at chance level, confirming that our algorithm's information gain comes solely from EEG responses. For ranking, results are compared to random guessing. For $L = 60$ candidates, the average target rank is 30, and the expected $d$ of a top-ranked face is 23.08.

**Data and distance ablations.** We evaluated our algorithm with varying dataset sizes ($N$) from 100 to 9234 to assess performance impact. Each dataset size was produced via uniform random sampling in $\Gamma_N$ without replacement. We also tested sensitivity to near-target stimuli by removing stimuli-response pairs below distance thresholds from the target face (0 to 40), reducing dataset sizes from 9234 to 806. We ran parallel experiments with uniformly down-sampled datasets matching each threshold's size to control for dataset size effects. Further details can be found in Appendix C.1.

**Alternative EEG representations.** To follow the representation learning with linear model evaluation framework [79, 51], we re-ran most experiments using EEGNet embeddings of the raw EEG responses, which has shown state-of-the-art performance in many tasks [40]. EEGPT [77] is a

more recent, promising, self-supervised pre-trained EEG representation but was published after our experiments had ended. While our contribution lies not in representation learning but in an SC-BCI framework that can be applied to any representation, future research could focus on comparing performance between various EEG representations. Additionally, we leveraged alternative representations to ensure open reproducibility, employing EEGNet to bypass limitations on sharing our original dataset imposed by privacy constraints. This enables the sharing of encoded data openly. An EEGNet model from Braindecode [63] was trained on a separate dataset specifically for task-relevant P3 classification. Further details and EEGNet results are in Appendix D.

### 6.4 Human validation experiments

We conducted two user studies to ground our results against human perceptual judgments. First, via **H-Rank**, where a user manually scored image similarity to compare the algorithm's scores with fine-grained human perception. Second, via **H-ID**, where a larger cohort determined the distance at which humans stop perceiving differences between faces. This second study contextualized all downstream experiments without requiring individual experiments for each scenario. Participants and precise experimental details can be found in Appendix C.2.

## 7 Results

**How algorithm and human scores correlate with the similarity function.** Figure 2 (left) shows that, in a randomly selected experimental run, CURSOR scores exhibit a strong negative correlation with the distance between the hypothesis and target faces, showing a Pearson coefficient $R$ of -0.82 ($p < 0.001$). These scores also align closely with those manually assigned by human evaluators, achieving an $R$ of -0.97 ($p < 0.001$). This outcome confirms that CURSOR and *H-Rank* scores match effectively, correlating well with our similarity metric.

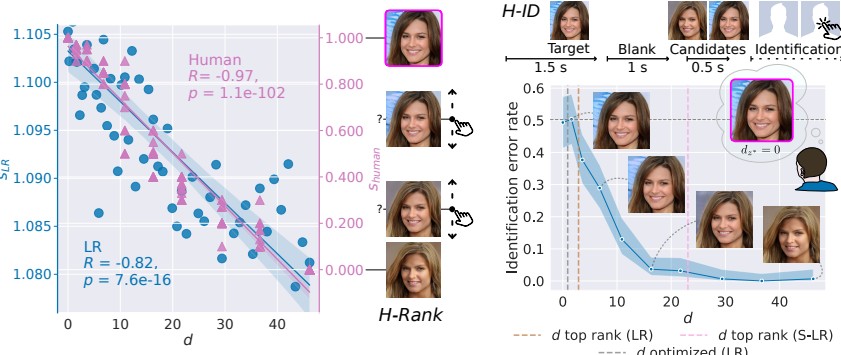

Figure 2: (**Left**) CURSOR and human scores correlate strongly with similarity. [Blue] Predicted scores ($S$) (Alg. 1 with LR estimator using full dataset) against ground-truth similarity ($d$) for 60 face images with regression lines and 95% CI. [Pink] Human-assigned *H-Rank* scores to sets of 10 facial images morphing away from the target [right axis]. (**Right**) A face at $d \leq 1.6$ has an error rate at chance level, suggesting perceptual indistinguishability to the target. A stimulus at $d \geq 16.3$ has an error rate nearing 0, confirming perceivable differences from the target. These statistics (Wilson Score shading at 95% CI) were collected during *H-ID* trials [top]. The vertical dotted lines show ranking and optimization performance, both recovering faces indistinguishable from the target.

**How human identification rate varies with image distances.** *H-ID* results (Figure 2 — right) confirmed near-chance identification at distances close to zero with $\eta(0) = 0.493$ (95% CI: 0.414, 0.573) and $\eta(1.6) = 0.503$ (95% CI: 0.425, 0.581) indicating perceptual indistinguishability for $d \leq 1.6$. At $d \geq 16.3$, faces became clearly distinguishable with $\eta(16.3) = 0.037$ (95% CI: 0.018, 0.074).

**How unsupervised ranking of images match their similarity to the mental target.** The predicted scores were utilized to rank hypotheses by their similarity to the mental target, with the Linear Regressor (LR) performing best across all metrics (Figure 3 and Table 1). The average correlation coefficient between predicted scores and ground-truth distances was -0.77 ± 0.04 SD for LR,

indicating that predicted scores effectively rank the hypotheses, with a mean top rank of 6.64 out of 60. At the top rank, our method achieved a mean distance of $2.9 \pm 3.08$ SD to the unknown target face, a distance only slightly perceptible to human subjects $\eta(3.6) = 0.377$ (95% CI: 0.304, 0.455).

Baseline performance was significantly worse than LR across all metrics (Mann-Whitney U, $p < 0.001$, Bonferroni corrected), with a significant perceptual difference of top ranked faces to the target (Fisher exact test comparing $\eta(3.6)$ and $\eta(21.7)$, odds ratio = 18.49, $p < 0.001$), confirming that EEG responses are the source of information and CURSOR unsupervised scoring function can convincingly rank hypothetical targets. *Additional metrics reported in Appendix E.3 and E.4, including top-k accuracies and a signal-to-noise analysis between methods.*

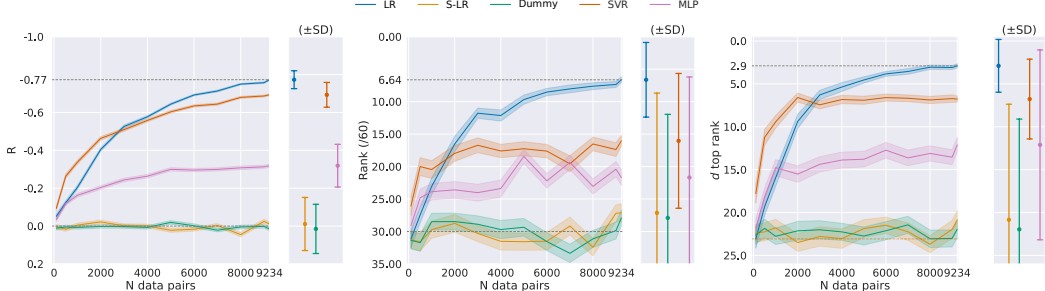

Figure 3: Rank-related performance for different estimators and data sizes (mean $\pm$ ste, std at $N$, inv. y-axis). (**Left**) Correlation coefficient ($R$) between scores and ground-truth distances. (**Middle**) Target rank out of 60. (**Right**) Euclidean distance to the target for top ranked candidate. The bottom dotted line is theoretical random performance. With LR rankings, human subjects could barely distinguish top-ranked faces from the target face with $\eta(3.6) = 0.377$ (95% CI: 0.304, 0.455).

Table 1: LR scores best across metrics, with LR, MLP, and SVR metrics significantly better than S-LR and Dummy controls (Mann-Whitney U, $p < 0.001$, Bonferroni corrected). The table shows ranking metrics (mean $\pm$ std) of different estimators and baselines at $N = 9234$ in Figure 3.

| Model | R | Rank | $d$ top rank |
|---|---|---|---|
| **LR** | **-0.77 $\pm$ 0.04** | **6.63 $\pm$ 5.75** | **2.90 $\pm$ 3.08** |
| S-LR | -0.01 $\pm$ 0.14 | 27.13 $\pm$ 18.45 | 20.85 $\pm$ 13.48 |
| Dummy | 0.01 $\pm$ 0.13 | 27.91 $\pm$ 15.95 | 21.97 $\pm$ 12.86 |
| MLP | -0.32 $\pm$ 0.11 | 21.68 $\pm$ 15.48 | 12.11 $\pm$ 11.07 |
| SVR | -0.69 $\pm$ 0.06 | 16.04 $\pm$ 10.38 | 6.77 $\pm$ 4.65 |

**How ranking performance varies across estimators and data sizes.** The performance of comparison estimators, SVR and MLP, was consistently lower than the Linear Regressor but above the control conditions, confirming that our algorithm works as long as the estimator used can improve prediction performance over shuffled versions of the dataset. In addition, Appendix Table 2 shows that a modern neural network embedding layer (EEGNet) followed by a fully connected network (BestMLP) performs at a level equivalent to a Linear Regressor. We further investigated estimators' performance on labeled data versus their performance on this task in more detail in Appendix E.2. Based on these results, and given that LR is faster to fit, we used only LR for the optimization tasks along with our two baselines (S-LR and Dummy).

**How optimization performs in dimensionally reduced spaces.** The optimization process successfully converged to an optimal face (Figure 4), positioned at an average Euclidean distance of 0.93 from the unknown target face in the original latent space — a difference barely perceptible to human subjects with $\eta(1.6) = 0.503$ (95% CI: 0.425, 0.581). In contrast, the control methods S-LR and Dummy failed to converge, yielding faces at average Euclidean distances of 24.09 and 22.51 from the target face, which are clearly perceptible to human subjects with $\eta(21.7) = 0.014$ (95% CI: 0.072, 0.032).

Baselines performance is significantly worse than LR (Mann-Whitney U, $p < 0.001$, Bonferroni corrected), with a significant perceptual difference (Fisher exact test comparing $\eta(1.6)$ and $\eta(21.7)$, odds ratio = 31.00, $p < 0.001$), confirming that EEG responses are the source of information and CURSOR unsupervised scoring function can successfully guide an optimization process.

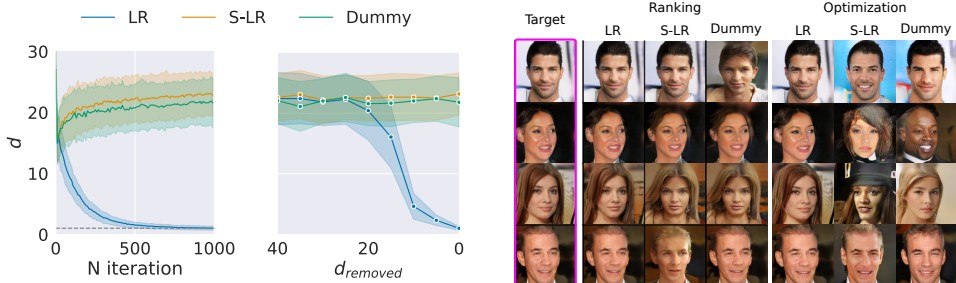

Figure 4: (**Left**) Optimization converges to $< 1$ distance to the target. Candidates distance ($d$) to the target against iteration number (left, mean $\pm$ std). We can recover near-target faces even when near-target stimuli are absent in the dataset, up to $d_{removed} = 10$. Best candidates found after 1000 iterations shown against ablation distances $d_{removed}$ around the target (right, mean $\pm$ std). (**Right**) Top-ranked (middle) and optimized (right) images per estimators from one randomly selected run and four randomly selected target faces (left). Images from CURSOR using Linear Regression (LR) are indistinguishable from the target image, while controls S-LR and Dummy exhibit visual differences.

**Visual inspection.**    Figure 4 (right) shows top-ranked and optimized faces for randomly selected runs and target faces between our best estimator (LR) and controls. Images for all 17 targets are included in Appendix Figure 9.

**Recovering ground-truth labels.**    Once $\hat{z}$ has been recovered in an unsupervised manner, we can reconstruct each distance via $\hat{d}_i = \rho_{\hat{z}}(z_i)$ for all $z_i \in \Psi_N$. This recovery step is significant because it serves as a proxy for calibration performance across any potential downstream supervised tasks and methods. In other words, if self-calibration can accurately reconstruct the ground-truth labels, then any subsequent supervised model can operate on these calibrated representations as if the data had been originally labeled. Our label-free CURSOR pipeline achieves reconstruction accuracy of RMSE $= 0.18 \pm 0.08$ (Appendix E.7). For context, a supervised decoder applied to the same EEG task and feature space (29 channels $\times$ 7 time windows, 203 features) reports RMSE $= 0.17 \pm 0.09$ in [71], despite minor differences in cross-validation meta-parameters.

## 8   Discussion, Limitations, and Conclusions

We addressed the problem of recovering image targets that a participant has in mind using only unlabeled EEG responses recorded while viewing a sequence of unlabeled images in continuous stimuli spaces. We demonstrated that our CURSOR algorithm can address this problem, recovering images indistinguishable from the targets participants had in mind, as confirmed by user studies.

**Dataset contribution.** To our knowledge, we release the first EEG dataset ($N = 9234$) for exploring SC-BCI in continuous domains with stimuli represented in a latent space and participants actively engaging with a target in mind.

**Limitations.** This study is restricted to face stimuli from one pre-trained GAN. Therefore, our experiments were limited to synthetic faces and generalization to real existing faces is yet to be explored. The acquisition protocol never explored latent distances $> 40$ (Euclidean) from the target, and relied on a predefined similarity function in the stimuli latent space. The optimization was run in a lower-dimensional latent subspace chosen for computational efficiency using simple heuristics, which may bias optimization. There may be alternatives that can improve our results. Real-time performance and cross-subject generalization remain untested.

**Broader impact.** Ethical considerations are critical as inferring intent without explicit consent raises privacy concerns [14, 17]. Working on explainable SC-BCIs and methods to assess confidence in their predictions will play a crucial role. The technology could broaden access to communication for users who cannot provide reliable overt feedback (e.g., individuals with severe motor impairments) and open new forms of adaptive human-computer interaction. Future work should pursue such benefits while remaining vigilant about misuse.

*Further limitations and ethical considerations are discussed in detail in the Appendix F.*

## Acknowledgements

Jonathan Grizou conducted this work during his tenure as an Assistant Professor at the University of Glasgow and subsequently through GrizAI Ltd. We gratefully acknowledge the financial support of both organizations.

The research was also partially funded by the the Alfred Kordelin Foundation (grant 230099) and the Finnish Foundation for Technology Promotion (grant 10168). We thank Roderick Murray-Smith helpful advice and access to facilities, and Huawei Cloud R&D Finland for travel support. Computing and storage resources were provided by the Finnish Computing Competence Infrastructure (FCCI; HILE ERC grant ILLUMINATOR, 101114623). We thank Jaakko Lehtinen, Tero Karras, Samuli Laine, and Timo Aila of NVIDIA for providing assistance and advice on GANs.

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

# Appendix

## A  Neurophysiological Data Acquisition

**Participants.**  The Ethical Committee of the University of Helsinki approved the study protocol, and participation in the neurophysiological experiment was advertised to a university population. In total, 31 participants volunteered for the study, aged $23.5 \pm 3.5$ (mean $\pm$ SD), from which 13 self-identified as male, 18 as female, and none as diverse. All participants were informed of the study's purpose and provided written consent before participation. After completing the study, participants were compensated with cinema vouchers.

**RSVP task**  Subsequently, the Rapid Serial Visual Presentation (RSVP) block started, presenting one facial image at a time every 500 ms using E-Prime 3 [68]. To minimize visual variance unrelated to facial features, a gray ellipsoid mask of $746 \times 980$ pixels was applied to each image. Each trial included 70 images: 28.5% related images and 71.5% unrelated images, where the imbalance is intended to enable the "oddball" paradigm. This design avoids confounds arising from participants attempting to predict the following image. It also facilitates the processing of data into overlapping epochs, therefore balancing acquisition speed with data quality [16]. The order of the images was randomized for each trial but ensuring that each presentation of a related image was followed by 1 to 8 unrelated images. In total, each participant completed 17 trials, therefore displaying 1120 images generated from the 17 image targets. Data acquisition sessions, including setup and EEG recording, lasted approximately one hour.

The participants received the following instructions before starting the task:

```
In this block you will be asked to concentrate and
mentally count every time you see this person below.
[Source image shown: young dark-haired female]
[Participant clicks the image]

That means you should ignore people who are not this person.
For example, ignore people such as this:
[Flanker image shown: young blond female,
different identity than source]
[Participant clicks the image]

Pictures of persons will be shown.
If the person is the one now shown in the middle, count it.
If the person looks different
(e.g. like the ones on the side), ignore it.
Click on the middle picture to begin!
[Three images shown: flanker, source, flanker;
same respective identities as before]
[Participant clicks the central image]

You will be shown faces.
Silently count the relevant ones.
Press SPACE to begin

[RSVP begins]
[Three images: flanker (constant),
changing image, flanker (constant)]
[70 central images shown]

How many did you count?
You will receive a prize if your
count is closest to the correct answer!
[Text input field]
[Participant types the count]
```

```
Thank you for participating and goodbye!
The experiment will close in 4 seconds.
```

**EEG recording.**    The experimental setup used a 32-channel EasyCap system, utilizing Ag/AgCl electrodes in the 10-20 system placed at FP1, FP2, F7, F3, Fz, F4, F8, FT9, FC5, FC1, FC2, FC6, FT10, T7, C3, Cz, C4, T8, TP9, CP5, CP1, CP2, CP6, TP10, P7, P3, Pz, P4, P8, O1, O2, and Iz, with AFz as ground. The participants seated at approximately 60 cm away from a 24" LCD screen with a resolution of $1920 \times 1080$ pixels and a refresh rate of 60 Hz.

The EEG responses were captured utilizing a BrainProducts QuickAmp USB amplifier, with voltages digitized via BrainVision Recorder at a sampling rate of 1000 Hz and using 0.01 Hz high-pass filter. During recording, signals were re-referenced to the common average. Electrooculographic (EOG) signals were recorded using bipolar electrodes, positioned in the peripheral area surrounding both eyes. Specifically, one pair was placed laterally to both eyes, while another pair was positioned above and below the right eye. The resulting signals are a time series of voltages distributed across channels in an $e \times c \times t$ tensor (EEG events, 32 channels, time).

**EEG filtering and artifact rejection.**    EEG responses were processed offline using MNE for Python [21]. Signals were filtered using a 0.1 Hz high-pass filter and a 50 Hz low-pass filter. Then, the segmented into time-locked bands, or *epochs*, each lasting 1 s and starting 0.2 s prior to the onset of each image stimulus [16]. Epochs were then baseline-corrected by subtracting the mean voltage per channel.

Artifact rejection was performed by eliminating epochs with a voltage exceeding $\pm 400$ μ in channels F3, Fz, F4, FC1, FC2, C3, Cz, C4, CP1, CP2, P3, Pz, and P4. Following, they applied Independent Component Analysis (ICA) to remove further artifacts such as eye movements. For this, they employed Autoreject [33, 34], resulting in a 5.3% epoch rejection rate. To expedite downstream computations, EEG responses were finally resampled to 250 Hz. Unlike its default behavior, Autoreject was configured to drop defective epochs rather than repair them. Channels with data corruption were dropped across all participants, reducing the number of channels from 32 to 29.

The resulting $e \times c \times t$ tensor (EEG events, 29 channels, time) was then standardized to zero mean and unit variance across trials for each participant, and across participants, to reduce inter-trial and inter-participant variability. EEG was windowed into seven equidistant time windows between 50 ms and 800 ms post-stimulus onset to streamline further and denoise the data. This yields an $e \times c \times w$ (EEG events, 29 channels, 7 time windows) tensor (i.e. $29 \times 7 = 203$ features per EEG event). CURSOR used flattened representations of the resulting EEG tensor, concatenated with image embeddings.

# B    Optimizing with Dimensionality Reduction.

The selection of appropriate dimensionality for PCA reduction was guided by the analysis of Pearson correlation statistics across varying numbers of principal components. Figures 5a and 5b illustrate the justification for our choices of 20 and 10 dimensions for EEG and face embedding reduction, respectively.

**EEG Dimensionality Reduction.**    Figure 5a displays the relationship between the number of PCA components and the Pearson correlation statistic in ranking experiments. Linear Regression method demonstrates gradual improvement that plateaus around 20 components. This asymptotic behavior suggests that 20 dimensions capture the majority of the relevant variance in the EEG data.

**Face Embedding Dimensionality Reduction.**    Similarly, Figure 5b presents the PCA analysis for face embeddings. Linear Regression method demonstrates gradual improvement that plateaus around 10 components. This indicates that a 10-dimensional representation is sufficient to retain most of the informative features of the face embeddings.

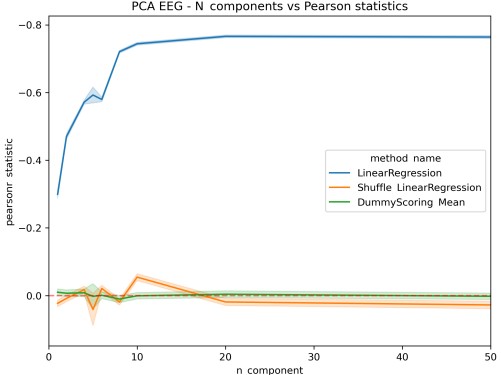
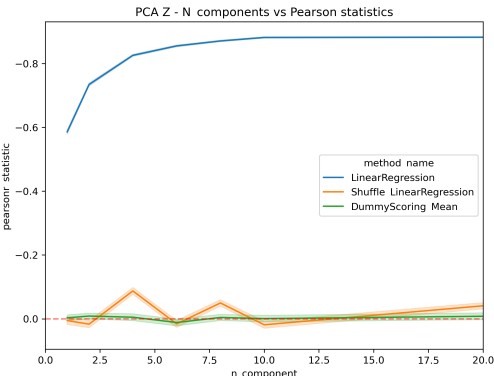

(a) Dimensionality reduction of EEG via PCA. The Linear Regression method reaches near-asymptotic performance at 20 components on our correlation metric.

(b) Dimensionality reduction of face embeddings via PCA. With EEG reduced to 20 D, the Linear Regression method reaches near-asymptotic performance at 10 components.

Figure 5: PCA component sweeps for (Left) EEG signals and (Right) face embeddings.

In both cases, the control estimators (Dummy and S-LR) consistently showing little to no correlation across all component numbers, hence no leakage of information despite significant dimensionality reduction.

These analyses support our decision to reduce EEG data to 20 dimensions and face embeddings to 10 dimensions, as these choices balance dimensionality reduction with preservation of relevant information, as evidenced by the near-asymptotic performance in correlation metrics.

## C  Experiments

### C.1  Comparative and ablation studies

**Estimators: Hyperparameters and Search.**  We used default estimators' meta-parameters with SVR using the radial basis function (RBF) kernel, and the MLP consisting of 3 hidden layers with (100, 50, and 25) neurons and a ReLU activation function. The learning rate for the MLP was set to 0.001, and the Adam optimizer used betas set at 0.9 and 0.999.

Additionally, we conducted an exhaustive hyperparameter search using "GridSearchCV" in scikit-learn [56] to optimize our Multi-Layer Perceptron (Best MLP) and SVR (Best SVR) models.

For the MLP estimator, we explored activation functions (Identity, ReLU), regularization parameter $\alpha$ (0.0001, 0.01, 0.1, 1), hidden layer architectures ranging from simple (100), to comparatively more complex (1000, ..., 1000) up to 10 layers, and adaptive learning rates for all experiments. Models were evaluated using cross-validated mean RMSE scores.

For the SVR model, we explored the regularization parameter $C$ (0.001, 0.01, 0.1, 1), kernel coefficient $\gamma$ ("scale", "auto"), and kernel type (linear, RBF). In total, 16 combinations were evaluated.

**Data and distance ablations.**  We evaluated our algorithm performance with varying dataset sizes ($N$) of [9234, 9000, 8000, 7000, 6000, 5000, 4000, 3000, 2000, 1000, 500, 100] to evaluate how the model performance is impacted by the amount of observed data. Each dataset was down-sampled to its final size using uniform random sampling without replacement.

We tested sensitivity to near-target stimuli by removing stimuli-response pairs below various distance thresholds from the target face [0, 5, 10, 15, 20, 25, 30, 35, 40]. This reduced dataset size to, respectively, [9234, 6644, 5432, 4335, 3413, 2668, 2047, 1452, 806]. To control for dataset size effects, we ran parallel experiments with uniformly down-sampled datasets matching each threshold's size.

**Optimization ablation and control.** We conducted this ablation studies to investigate the importance of near-target stimuli on CURSOR's performance. Because this removed data from the stimuli-response set, our experiment also impacted data scarcity. To control for dataset size effects, we ran control experiments with uniformly down-sampled datasets matching each near-target ablation threshold's size. Figure 6 illustrates the results of these experiments (mean ± SD, x-axis inverted).

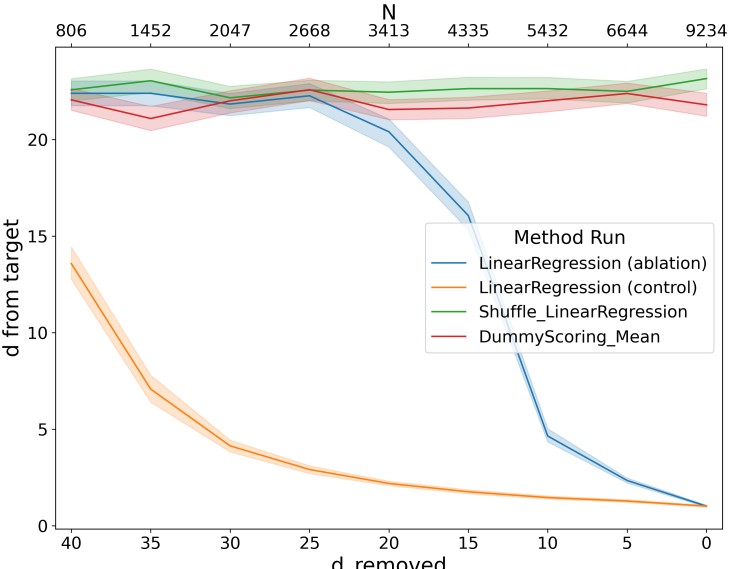

Figure 6: Comparison of CURSOR performance under near-target ablation and uniform data reduction conditions. Figure shows distance to target of candidates found after 1000 iterations against dataset ablated within specified distances $d_{removed}$ around the target (mean ± SD). We can recover near-target faces even when near-target stimuli are absent in the dataset, up to $d_{removed} = 10$, demonstrating CURSOR's ability to infer beyond its observation space and its robustness to data scarcity. The disparity in performance between ablation and control conditions at comparable $N$ values highlights the importance of near-target stimuli for CURSOR's accuracy.

**Estimator performance on supervised tasks vs CURSOR Performance.** We trained various estimators on ground truth data to estimate their theoretical capacity for predicting $d$ from $e$. The root mean square error (RMSE) is used as a performance metric, with lower values indicating better performance. The primary objective of this study is to investigate whether CURSOR's effectiveness is dependent on having a highly accurate estimator, and to explore the relationship between the estimator's capacity to capture data relationships and its performance within the CURSOR framework.

## C.2 Human validation experiments

***H-Rank* — Ranking face similarity with unlimited time.** The *H-Rank* experiment evaluates the ability to manually assign distance scores and rank images from single latent trajectories between the target and the farthest image used in the EEG experiment. An evaluator, uninvolved in the EEG experiment and sourced from a university population, was presented with sets of 10 progressively morphed images spanning distances from 0 to 40.16 ($D = \{0, 1.6, 3.6, 6.8, 10.9, 16.3, 21.7, 29.4, 36.7, 46.2\}$ (Figure 2 right) and the scale was normalized between 0 and 1 for ease of evaluation. The remaining 8 images were unlabeled and displayed in a randomized order. The task was not timed, and all 10 images per set were displayed simultaneously. In total, the evaluator assessed 170 images, corresponding to 10 images per each of the 17 targets. An alternative representation of the results from the main paper is shown in Figure 7.

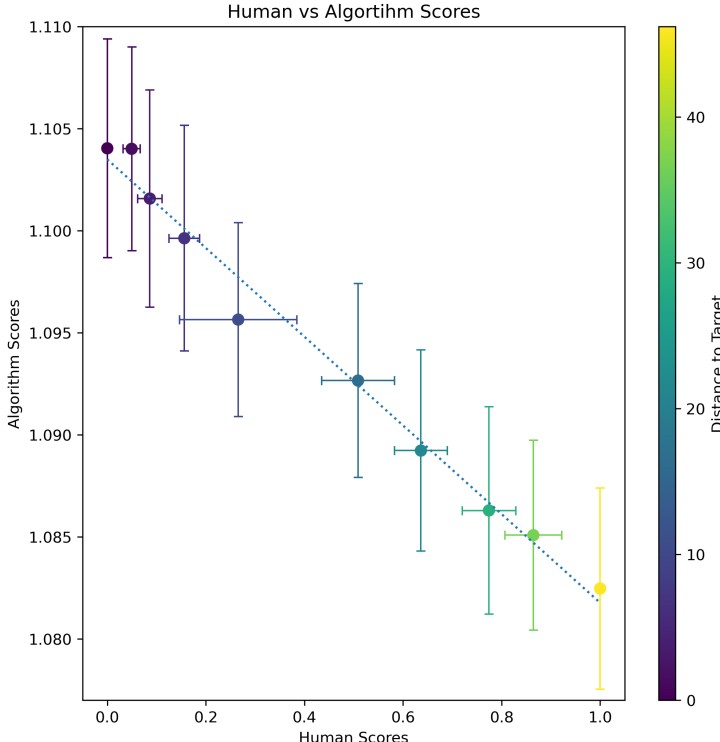

Figure 7: Human scores attached to a subset of test faces (unlimited time) vs Algorithm score from our method using Linear Regressor on full dataset (error bars show the standard deviation). Each dot represents a test face colored by the distance this face is from the target face. Our algorithms scores correlates strongly with human scores.

***H-ID — Face identification from short exposure.*** The *H-ID* experiment aims to contextualise our results by determining the distances in $Z$ at which two faces are indistinguishable or become clearly differentiable by human evaluators.

During *H-ID* trials, a target face $z^*$ was presented centrally for 1.5 s, followed by a 1-second blank interval. Then, two faces were displayed side-by-side for 0.5 s: the target face $z^*$ and a comparison face $z_c$ at a distance $d_{z_c} = \rho_{z^*}(z_c)$ from the target. The 0.5 s exposure duration was chosen to match the stimulus presentation time used to acquire the dataset in our neurophysiological experiments. Participants were instructed to click on the face they believed matched the target face shown previously.

For each trial, the target face was selected randomly from the 17 target faces available in our dataset. The position of the target face in the selection stage (left or right) was randomized across trials. The distance $d_{z_c}$ between the target and comparison face was randomly chosen within the $D$ set as for *H-Rank*. Each participants completed 30 such trials. A local machine was used in the setup, using Flask as backend and HTML/CSS/JS for the frontend.

53 participants volunteered for the study, aged between 20 and 39 years (mean = 24.5, SD = 3), of which 8 self-identified female, 45 as male, and none as diverse. All participants were informed of the study's purpose, reported no visual impairments, and provided written consent before participation.

We calculated the identification error rate $\eta(d_{z_c})$ as the proportion of incorrect selections at each distance $d_{z_c}$, using Wilson score for confidence intervals. At $d_{z_c} = 0$, we expect chance performance (error rate 0.5), while for clearly differentiable faces, the error rate should approach 0. *H-ID* was designed to reveal the transition between these extremes.

## C.3 Computational environment

The computations were performed locally on GNU/Linux and MacOS-based machines:

- A Lenovo Thinkpad t480 running Ubuntu 22.04, an i5-7300U Intel CPU at a base frequency of 2.6 GHz, Intel HD Graphics 620, and 32GB / 2400MHz DDR4 RAM;

- A custom machine running Linux 6.10.3, an AMD Ryzen 7 7840U 16 core at a base frequency of 3.3 GHz with Radeon 780M Graphics, and 64 GB / DDR5-5600 RAM;

- A MacBookPro running MacOS Sonoma 14.3, an Apple M1 Max with 10 cores and 64GB RAM.

Most computationally intensive tasks were executed on an on-premises AlmaLinux 8.7 cluster with a SLURM scheduler and AMD EPYC 7452 CPUs. Computations utilized embarrassingly parallel tasks, each running on a single CPU core. Jobs were run on nodes designated for "short" or "medium" tasks, suggesting relatively low HPC resource demands per job. We launched one job array per estimator due to varying resource requirements. Initial computational load was assessed with a job configuration of 1 CPU core, 800 MB of RAM, and a walltime of 1 hour, and adjusted based on observed resource usage. Maximum RAM usage and walltime, with at least a 20% buffer for variability, were 400 MB and 4 hours for Linear Regression, 400 MB and 4 hours for MLP, and 600 MB and 15 hours for SVR.

There are no specific package version requirements. We use Python 3.11, the oldest stable version at submission time supporting advanced enumerations for quality of life improvements, and therefore dependency versions are flexible. Details on the packages can be found in the dependencies file in the code.

# D    Alternative EEG Representations

EEGNet model was trained on a separate dataset specifically for visual saliency detection using P3-based binary *classification*. The EEG data was collected using the same equipment and GAN as in the main experiment to minimize confounds, drift, and noise differences.

## D.1    Neurophysiological experiment for visual saliency in faces

After obtaining approval from the Ethical Committee of the University of Helsinki, we recruited 31 participants from a University population. Images were generated using the same GAN architecture trained on a dataset of celebrity faces as in the main experiment, sampled from a 512-dimensional multivariate normal distribution. These images were manually categorized into eight visual categories: blond, dark-haired, female, male, old, young, smiling, and non-smiling. The study apparatus, including the EEG recording setup and stimuli presentation, was consistent with the main experiment.

After providing informed consent, participants were instructed to identify whether each face shown in a rapid serial visual presentation (RSVP) sequence of 70 images contained one of the eight salient features (*target*) while maintaining a mental count. Each salient feature was presented in 4 RSVP trials before switching to the next category. The order of target and non-target images was randomized, as in the main experiment.

Following data collection, EEG data was preprocessed offline using a high-pass filter at 0.2 Hz and a low-pass filter at 35 Hz. The data was then epoched and baselined as in the main experiment, with artifacts removed using a threshold of $\pm 400$ µV based on the highest absolute maximum voltage. This preprocessing step removed two participants due to artifacts and approximately 11% of epochs, resulting in a dataset with an average of 3251 epochs per participant.

## D.2    Learning task and prediction setup

EEGNet models were trained to classify the presence or absence of salient features in face images. Each model utilized EEG epochs containing 29 channels and 376 time points as input, with a binary classification label as the output. The training process spanned 10 epochs with a batch size of 32, employing a binary cross-entropy loss and the Adam optimizer with a learning rate of 0.001 and moment betas set to 0.9 and 0.999. For evaluation, data corresponding to one salient visual category was left out during training. After training, the final classification layer of the EEGNet model was removed, and the remaining network was used to generate embeddings using EEG data from the main experiment as input.

# E Results

## E.1 Comparative and ablation studies

**Estimators: Hyperparameter search.** We ran hyperparameter optimization to identify the best-performing MLP and SVR models for the CURSOR framework ("Best MLP" and "Best SVR").

For the Best MLP model, identity activation generally outperformed ReLU, with moderate alpha values (0.1) yielding better results. Simpler architectures (2–3 layers) often performed comparably to or better than more complex ones. The best-performing configuration consisted of identity activation, alpha = 0.1, and two hidden layers of size 100 with an adaptive learning rate. This configuration achieved a mean RMSE score of $0.9138 \pm 0.0142$ (mean $\pm$ SD).

For the Best SVR model, linear kernels outperformed RBF kernels. Performance was relatively stable across $C$ values for linear kernels, with the best performance at $C = 0.01$. The "scale" and "auto" $\gamma$ settings produced identical results for linear kernels. RBF kernels showed lower performance, suggesting linear separability in the feature space. The best-performing configuration was $C = 0.01$, $\gamma =$ "scale", and the linear kernel, achieving a mean RMSE score of $0.9279 \pm 0.0115$ (mean $\pm$ SD).

## E.2 Estimator performance on supervised tasks vs. CURSOR performance

**Aligned vs. Shuffled Performance.** Figure 8a presents a scatter plot comparing the RMSE of each estimator on aligned and shuffled datasets. Baseline methods exhibit equal performance on both datasets, as expected. Despite displaying the lowest absolute performance, the MLP shows relative improvement over the shuffled dataset, allowing it to outperform baselines with better absolute RMSE scores in the CURSOR framework. This finding suggests that CURSOR's effectiveness is not solely dependent on having the best possible estimator in absolute performance.

**RMSE vs. Pearson Correlation.** Figure 8b illustrates the relationship between estimator performance on ground truth data and the Pearson correlation statistic for the information retrieval (IR) task. The Best SVR model achieves a lower RMSE than the standard SVR but demonstrates lower performance in ranking correlation with ground truth distances. The MLP performs nearly as well as the Best SVR in the CURSOR framework despite a substantial difference in RMSE. This counterintuitive result is possible due to CURSOR's scoring function, which measures relative improvement. Further investigation into the mechanisms behind this phenomenon could yield valuable insights.

**Conclusions.** The relationship between estimator characteristics and CURSOR performance is complex, unlike typical supervised or unsupervised learning scenarios. These findings argue for ensemble methods that dynamically test different estimators within the CURSOR framework. Future research should focus on developing robust confidence metrics for CURSOR scores estimations.

In conclusion, this study reveals the complex interplay between estimator characteristics and CURSOR performance. The framework's ability to extract useful information from estimators with suboptimal absolute performance highlights its robustness and adaptability. These findings open avenues for future research, particularly in dynamic estimator selection strategies and improved confidence estimation techniques.

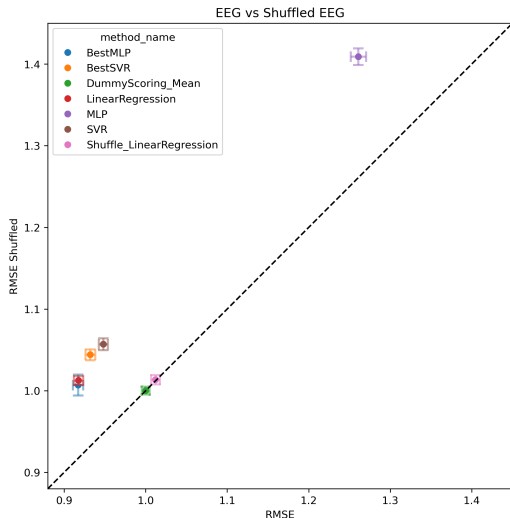
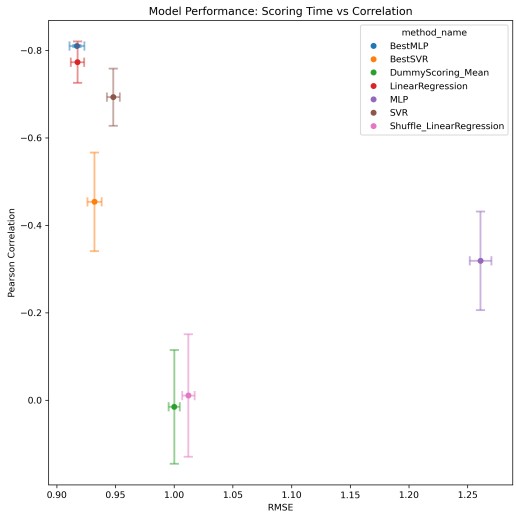

(a) Comparison of estimator performance on aligned vs. shuffled EEG data. The x-axis shows the RMSE on aligned data, while the y-axis shows the RMSE on shuffled data. Each point represents a different estimator, with error bars indicating standard deviation. The diagonal dashed line represents equal performance on both datasets. Points above this line indicate better performance on aligned data compared to shuffled data, which is desirable for CURSOR. Note that while some methods (e.g., MLP) may have higher absolute RMSE, they still show improvement over the shuffled baseline, demonstrating potential utility in the CURSOR framework.

(b) Relationship between estimator performance and CURSOR effectiveness. The x-axis shows the RMSE of each estimator on the target face dataset (lower is better), while the y-axis shows the Pearson correlation between the estimator's rankings and ground truth distances (higher absolute value is better). Each point represents a different estimator method, with error bars indicating standard deviation. Notably, methods with similar RMSE can have vastly different correlations, and some methods with higher RMSE (e.g., MLP) can achieve correlations comparable to methods with lower RMSE (e.g., BestSVR). This demonstrates that absolute estimator performance does not necessarily predict effectiveness within the CURSOR framework.

### E.3 Ranking images and optimizing the image embedding

Images ranked and generated for the 17 targets are shown in Figure 9. Table 2 extends the main paper's results for EEG preprocessing using windowing (left) and includes a subset re-run using EEGNet embeddings (right). Trends and magnitudes are aligned between the two EEG representations, suggesting EEGNet embeddings as an alternative preprocessing method in the CURSOR framework. This allows us to provide embeddings as attached data, overcoming the challenge of sharing raw EEG data.

| | **EEG windowing** | | | **EEGNet** | | |
|---|---|---|---|---|---|---|
| Regressor | $d$ top rank | R | Rank | $d$ top rank | R | Rank |
| LR | $2.90 \pm 3.08$ | $-0.77 \pm 0.04$ | $6.63 \pm 5.75$ | $2.66 \pm 3.02$ | $-0.76 \pm 0.05$ | $6.21 \pm 6.03$ |
| S-LR | $20.85 \pm 13.48$ | $-0.01 \pm 0.14$ | $27.13 \pm 18.45$ | $21.07 \pm 13.73$ | $-0.01 \pm 0.13$ | $26.35 \pm 16.93$ |
| Dummy | $21.97 \pm 12.86$ | $0.01 \pm 0.13$ | $27.91 \pm 15.95$ | $22.29 \pm 12.44$ | $-0.01 \pm 0.14$ | $27.73 \pm 16.60$ |
| MLP | $12.11 \pm 11.07$ | $-0.32 \pm 0.11$ | $21.68 \pm 15.48$ | $13.26 \pm 11.46$ | $-0.31 \pm 0.11$ | $19.85 \pm 15.01$ |
| BestMLP | $4.30 \pm 2.64$ | $-0.81 \pm 0.00$ | $9.67 \pm 10.02$ | $3.91 \pm 3.53$ | $-0.76 \pm 0.04$ | $6.14 \pm 5.89$ |
| SVR | $6.77 \pm 4.65$ | $-0.69 \pm 0.06$ | $16.04 \pm 10.38$ | $6.12 \pm 4.61$ | $-0.69 \pm 0.06$ | $15.65 \pm 12.85$ |
| BestSVR | $12.60 \pm 5.53$ | $-0.45 \pm 0.11$ | $43.34 \pm 13.37$ | $13.17 \pm 4.48$ | $-0.40 \pm 0.12$ | $44.86 \pm 13.88$ |

Table 2: Euclidean distance ($d$) at the top rank, coefficient of determination (R), and rank shown as mean $\pm$ standard deviation for each estimator: linear regression (LR), shuffled linear regression (S-LR), dummy scoring mean, default and optimized ("best") multi-layer perceptron (MLP), and default and best support vector regression (SVR). Results are presented for two alternative EEG representations: one applying windowing over the raw signal, and the other using EEGNet embeddings.

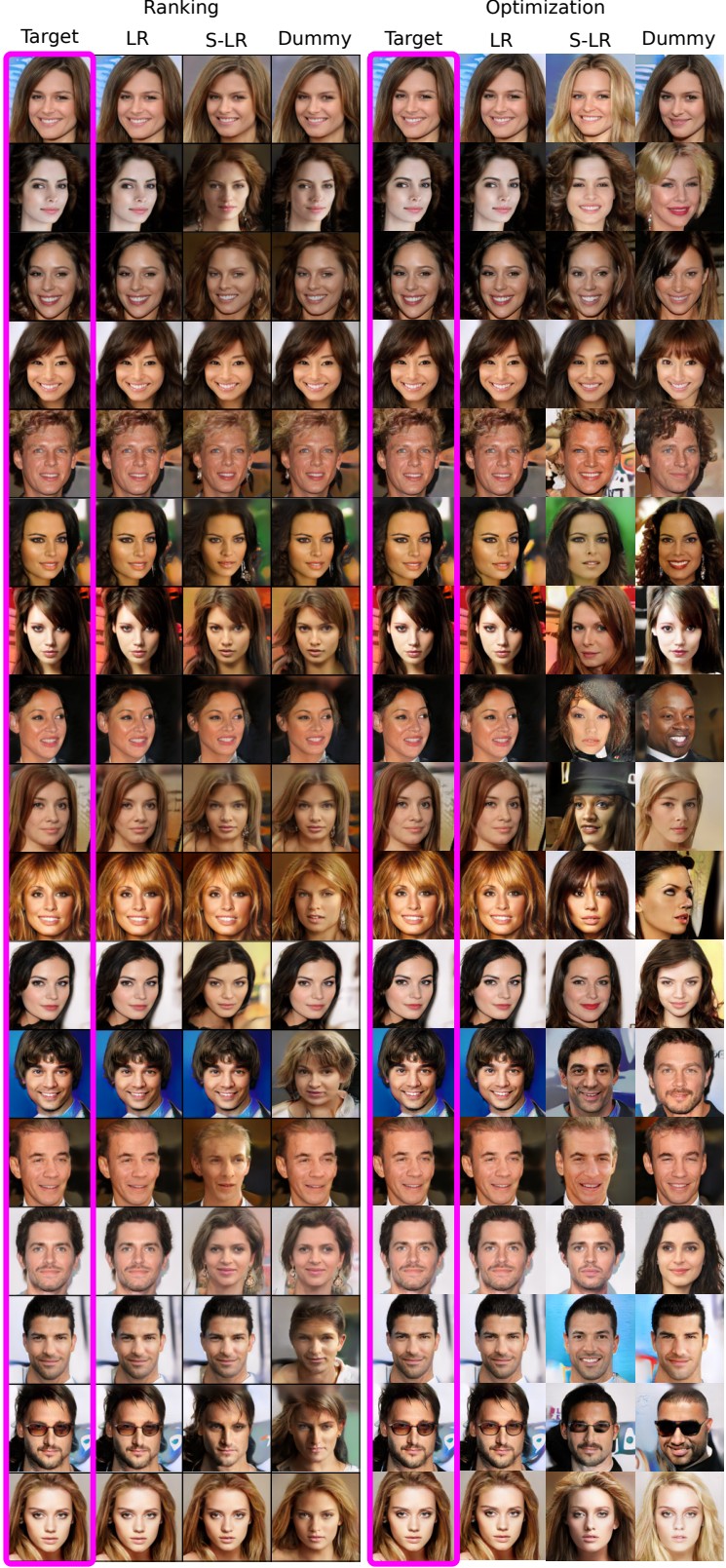

Figure 9: Images matching the top-scored hypothesis for each image target (left) and generated from the optimization (right) using Linear Regression (LR) and the control conditions Shuffle Linear Regression (S-LR) and Dummy Mean Scoring. Ideally, there would be no perceptible differences between LR and the target, while S-LR and Dummy would exhibit very high visual differences compared to the target.

### E.4 Additional retrieval metrics

The Top-$k$ and Min-d-at-Top-$k$ metrics' tables, namely Table 3 and Table 4 respectively, were not included in the main text as we considered the **distance-at-top-rank (d-at-top rank)** a more informative metric for our perceptual task, given that the rank does not necessarily reflect perceptual similarity.

Table 3: Top-$k$ Accuracies: Linear Regression (LR) vs. Theoretical Random Retrieval (60 Candidates)

| Model | Top-1 Acc | Top-3 Acc | Top-5 Acc | Top-10 Acc |
|---|---|---|---|---|
| LR | $0.171 \pm 0.029$ | $0.353 \pm 0.037$ | $0.565 \pm 0.038$ | $0.800 \pm 0.031$ |
| Random | $0.016 \pm 0.010$ | $0.049 \pm 0.016$ | $0.081 \pm 0.021$ | $0.155 \pm 0.028$ |

Table 4: Minimum Distance to Target in Top-$k$ Retrieved Hypotheses (Min-d-at-Top-$k$)

| Model | Top-1 Min Dist | Top-3 Min Dist | Top-5 Min Dist | Top-10 Min Dist |
|---|---|---|---|---|
| LR | $2.898 \pm 3.067$ | $1.010 \pm 1.312$ | $0.524 \pm 0.861$ | $0.176 \pm 0.450$ |
| Random | $23.064 \pm 13.316$ | $11.532 \pm 8.933$ | $7.688 \pm 6.498$ | $4.194 \pm 3.828$ |

In addition to the retrieval metrics presented above, we further investigated the stability and robustness of the CURSOR score with respect to random data splits and model variability. This analysis was motivated by reviewer feedback concerning the potential sensitivity of the CURSOR score—computed as a relative RMSE ratio—to shuffling and data partitioning randomness.

As a first step, we visualized the relationship between the predicted scores and the ground-truth perceptual distances across all folds and random splits (Figure 10). This plot shows the mean and standard deviation of each model's predictions as a function of the true perceptual distance, allowing us to inspect the uncertainty of scores directly. Notably, the MLP model displays visibly larger uncertainty ranges compared to other methods.

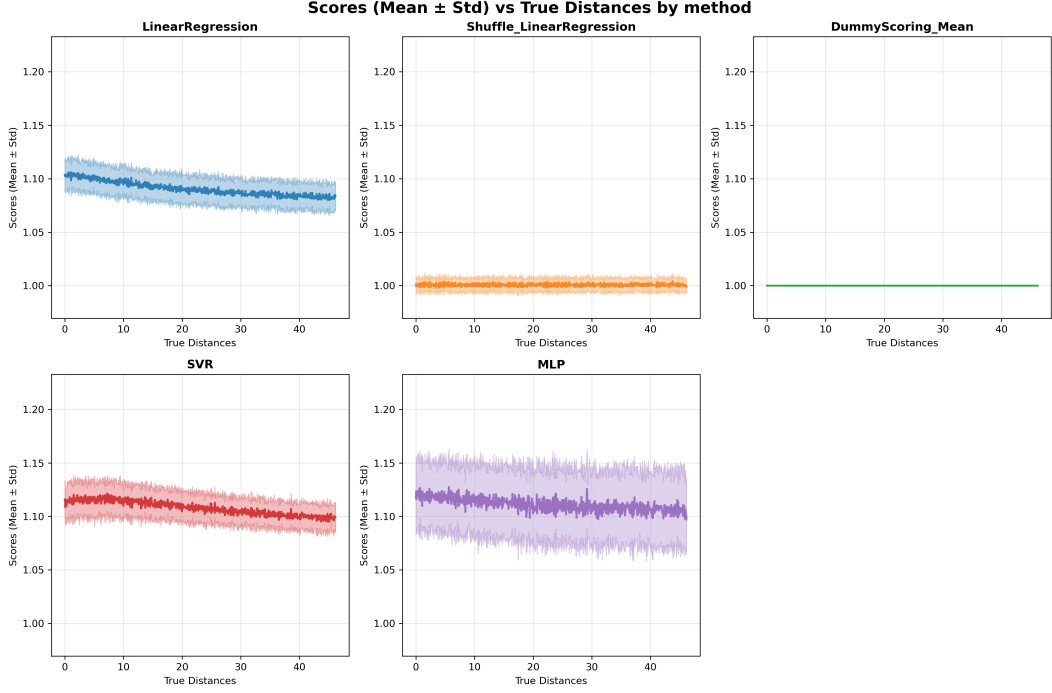

Figure 10: Relationship between mean $\pm$ standard deviation of predicted scores and ground-truth perceptual distances, aggregated over all runs. Each panel corresponds to a different estimator, illustrating how uncertainty varies across the perceptual space and highlighting the higher dispersion observed for the MLP model.

Across 10-fold cross-validation experiments, we confirmed those observations as the standard deviation of the CURSOR scores was approximately 2.5% of the mean for most methods at $N = 9234$, to the notable exception was the MLP model, which exhibited slightly higher variability at around 4% (Figure 11).

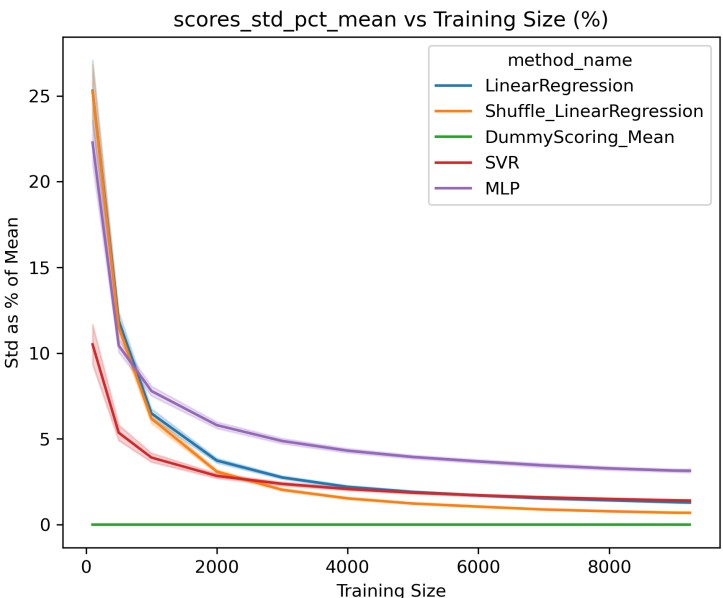

Figure 11: Evolution of the standard deviation as a percentage of the mean CURSOR scores across increasing $N$ for all evaluated methods. The plot shows an exponential decay trend from approximately 20% to 2.5% for most models (4% for MLP), indicating improving score stability with increasing dataset size.

Finally, to quantify robustness more systematically, we computed a signal-to-noise ratio (SNR)–like metric for the ranking task, defined as:

```
SNR = np.std(np.mean(scores, axis=1)) / np.mean(np.std(scores, axis=1))
```

where scores are matrices of shape $(60, 10)$ representing the ranking results across folds and random splits. This metric compares the signal variation across conditions to the typical uncertainty within them, and results are shown on Figure 12. Interestingly, for both S-LR and MLP, this ratio remained roughly constant around 0.4 for all $N$. In contrast, LR and SVR began at similar values but increased approximately linearly with $N$, reaching 0.6 and 0.52, respectively, at $N = 9234$.

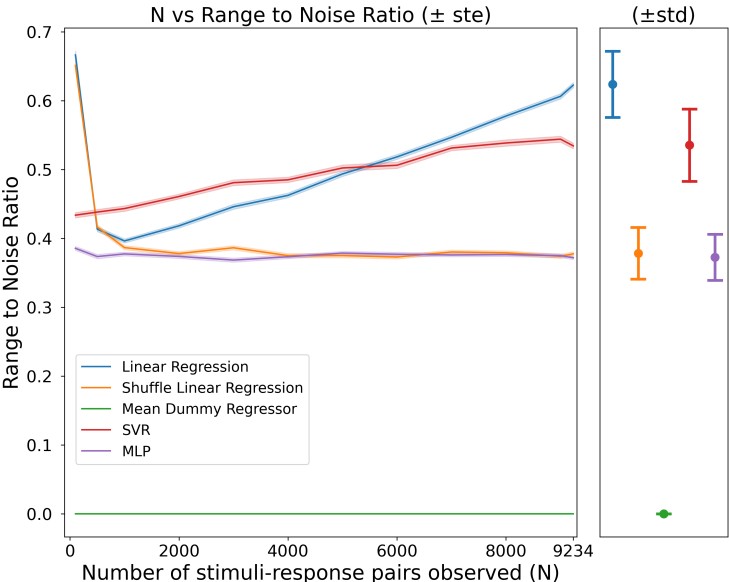

Figure 12: Signal-to-noise ratio (SNR) as a function of $N$ for each model. LR and SVR show a clear linear increase in SNR, reaching 0.6 and 0.52 at $N = 9234$, while S-LR and MLP remain approximately constant around 0.4 across all $N$.

When interpreted alongside the results in Figure 3, this SNR analysis provides a coherent explanation for the comparatively lower perceptual retrieval performance of MLP. Although MLP correlates with ground-truth distances, its higher cross-validation variance translates into a lower effective SNR and hence less reliable top-rank predictions.

## E.5 Computation time vs. performance

Figure 13 examines the relationship between computation time and performance. The primary observation is the marked variability in computation time among methods. Given the weak correlation between performance and computation time, it may be advantageous to focus on faster-to-train estimators when implementing CURSOR initially.

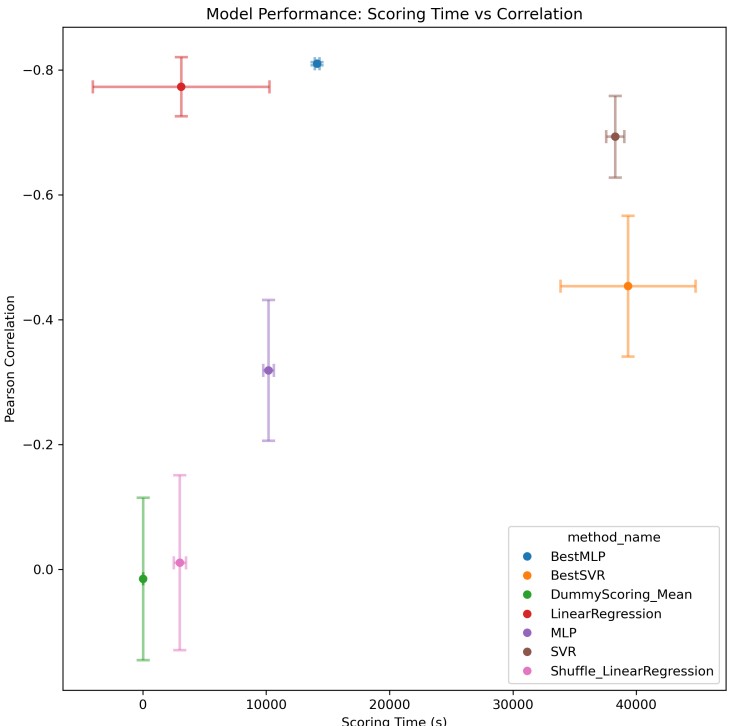

Figure 13: Trade-off between computational time and performance in the CURSOR framework. The x-axis shows the scoring time in seconds, while the y-axis shows the Pearson correlation between the estimator's rankings and the ground truth distances. Each point represents a different estimator, with error bars indicating standard deviation.

## E.6 Optimization ablation and control

Our primary finding is that CURSOR can recover a target face within $d < 5$ even when no stimuli within $d < 10$ have been observed. This demonstrates CURSOR's ability to infer beyond its current observation space, a crucial feature for practical applications where near-target stimuli may be scarce. It also suggests that a non-logarithmic sampling scheme might be viable for future self-calibrating experiments.

Comparing the ablation and control conditions reveals the significant impact of near-target stimuli on performance. When $d_{removed} > 10$, CURSOR was unable to recover a face at $d < 5$ from the target, even with $N = 5432$ stimuli-response pairs. In contrast, under uniform data reduction (control condition), similar performance was achievable with only $N = 2047$ pairs.

This performance disparity at comparable $N$ values confirms that responses to stimuli close to the target significantly enhance CURSOR's accuracy. It suggests that strategic sampling, informed by CURSOR scores could potentially improve the efficiency of the algorithm in practical scenarios and is worth further research. These results highlight CURSOR's robustness to data scarcity and its ability to perform well even with limited near-target information, while also emphasizing the value of near-target stimuli for optimal performance.

### E.7 Recovering ground-truth labels.

Using the best stimuli $\hat{h}$ identified during the optimization process, we reconstructed labels for each observed response $e_i$ as the distance $d_i^{\hat{h}} = \rho_{\hat{h}}(z_i)$. The quality of these reconstructed labels is measured by their RMSE compared to the ground-truth labels $d^{z^*}$. For LR, RMSE$(d^{z^*}, d^{\hat{h}})$ is $0.18 \pm 0.08$, representing less than 0.4% of the range of $d$ values and outperforming S-LR at $15.20 \pm 3.21$ and Dummy at $18.23 \pm 2.99$ (both Mann-Whitney U, $p < 0.001$, Bonferroni corrected). It demonstrates our method's ability to recover ground-truth labels without supervision and suggests potential effectiveness in downstream tasks requiring labels comparable to calibrated approaches.

## F  Discussion, Limitations and Conclusions

### F.1  Limitations.

**Online replication.**   Our findings should be replicated in online settings using protocols that match our simulations. This might surface additional constraints for practical deployment. For example, prolonged or repeated exposure might attenuate the P3 response, potentially affecting the system's reliability [1, 7].

**Real faces.**   Our experiments were limited to synthetic faces and generalization to real existing faces is yet to be explored and would constitute interesting future research. The core challenges would be: 1) new EEG experiments to acquire a novel dataset of stimuli-responses pairs, and 2) the introduction of a new component to the algorithm, which allows a quantitative distance definition that correlates with real faces' visual similarity. This could involve encoding the faces in an embedding space of a pre-trained model, or using predictions from a pre-trained contrastive model comparing hypothetical targets to observed stimulus, among others.

**Using more complex estimators.**   Although we have explored the combination of EEGNet embeddings and MLP estimator, we did not directly used of full deep neural network/transformer architecture, which might lead to better *absolute* prediction performance. However, as CURSOR computes a relative score, the absolute performance of an estimator on the ground truth data might not always be the most significant factor. Understanding this relationship would constitute interesting future research, which Appendix Figure 8a started to explore.

**Optimization in reduced dimensional space.**   While computational constraints motivated our choice to conduct optimization in a low-dimensional subspace (we run over 8000 experiments for our controls, as shown in Appendix Figure 6), it is considered standard to use dimensionality reduction as part of a black-box optimization procedures [12, 44, 29, 65].

Our decision to use 10D/20D is supported by Appendix Figure 5. We ran the ranking task for different EEG dimensions (using PCA). At 20D, we reached near-asymptotic performance. Then, maintaining EEG at 20D, we reduced the image space (via PCA), and 10D maintained similar near-asymptotic performance. Performance was measured once images were converted back to their original 512D to ensure apples-to-apples comparison with ranking and human studies. In other words, we used the smallest dimensionality that preserves task performance (20-D EEG, 10-D faces) consistent with low-effective-dimension assumptions in common frameworks [12, 15, 44].

The limitation is not in using dimensionality reduction per se, as all our metrics are reported after faces are projected back to their original 512D (using PCA inverse mapping) to ensure apples-to-apples comparison with ranking and human studies. Instead, it is because we relied on external metrics to choose 10D/20D, which would not be available in a real scenario (i.e., correlation of scores vs ground-truth on the ranking task).

Future research could study performance using internal dimensionality-reduction metrics instead (e.g., reconstruction metrics). We did not test such latent space variants in this work and there may be alternatives that can improve over our results.

**Active sampling.**   We have not explored any active sampling strategies and, for all experiments, samples where draw uniformly without replacement. Active sampling [11] might help reduce the

amount of EEG-stimuli pairs required to identify a target, which is currently a significant limitation to practical use.

**Meta Self-Calibration** In its current form, our self-calibration algorithm does not learn any "persistent" parameters (that are estimated and then frozen for later use), apart from those of its estimators $f(\theta_h)$ [(Alg. 1, line 3) and (Alg. 1, line 6)]. It would be interesting research to generalize a CURSOR estimator for a specific task/domain based on a large unlabeled dataset so that $f_\theta(\Gamma, h) \mapsto scores$ for any $\Gamma$ and $h$. In this scenario, $\theta$ would constitute learned "persistent" parameters for a generalized CURSOR estimator.

### F.2 Ethical implications and potential applications.

Ethical considerations are crucial in this line of research as the ability to infer people's intent without explicit questioning or consent raises important privacy and ethical concerns that warrant caution of application scenarios [14, 17]. Potential negative societal impacts, due to performance instability in risk-sensitive applications, need to be carefully addressed. Working on explainable SC-BCIs and methods to assess confidence of their predictions will play a crucial role.

Physiological data can be sensitive and re-identifiable, and consent mechanisms are often opaque in practice, which increases the importance of cognitive privacy aspects for BCI research [66, 52, 39, 14].

The core risk is that models or datasets are repurposed beyond the stated task to infer mental content without informed, ongoing consent. For that, we advocate for:

- Acknowledging risk explicitly: We will state that inferring internal mental content without user intent is unacceptable and a central ethical concern [10, 64].
- Requiring opt-in, revocable consent: Participation must be explicit, comprehensible, and withdrawable, with data erasure upon request [6, 49].
- Imposing purpose limitation and governance: Models and data may only be used for the approved task and context, with new consent required for secondary use and audits recorded [53, 54, 41].
- Implementing data minimization and secure handling: Prefer on-device or sandboxed processing; encrypt data in transit/at rest; set short retention windows; significantly limit the sharing of raw EEG [46, 57].
- Increasing transparency and accountability: Release model cards and datasheets documenting intended use, limits, failure modes, and oversight plans [48, 18].
- Centering on user agency: Provide a visible "record/mute" control, require explicit start/stop for any recording, and disable inference outside the study context by default [14].

Furthermore, our study is offline, task-bound, and conducted with consenting volunteers; we do not perform real-time or covert inference, and outputs are analyzed in aggregate, which serve as mitigations.

Despite potential risks, this technology could have beneficial applications, particularly in diagnostic scenarios where user feedback is unreliable or calibration is difficult, such as early diagnosis of neurodegenerative disorders [58]. Future research should explore such positive use cases while remaining vigilant about potential misuse.

