# OpenReview forum: "Self-Calibrating BCIs: Ranking and Recovery of Mental Targets Without Labels"
_NeurIPS.cc/2025/Conference — NeurIPS 2025 poster_

### Official Review · Reviewer_Dn3x · 2025-06-06

**Clarity:** 2
**Significance:** 3
**Originality:** 3
**Rating:** 4
**Confidence:** 3

**Summary:**

This paper presents a self-calibrating brain-computer interface approach to identify and reconstruct a mental image target (a face that the user is thinking of) using only unlabeled EEG signals and stimuli embeddings. Unlike prior approaches that rely on labeled data or pre-trained decoders, CURSOR estimates target similarity scores in a continuous latent space derived from a generative model, without requiring any supervision. The method defines a score function based on how well a regressor trained on EEG-to-distance pairs generalises compared to a shuffled baseline. Using this score, CURSOR ranks and optimises hypothesised mental targets in latent space. The authors also describe a large EEG dataset collected under controlled conditions and they validate their approach through correlation with human perceptual judgments, control experiments, ablations and benchmarks.

**Questions:**

Can you justify the need for optimisation in reduced-dimensional spaces beyond computational cost? Does optimisation in 10D/20D reliably reflect performance in the original 512D/203D latent spaces?

Line 159 (logarithmic sampling) and line 161 (“bias mitigation”) lack citations or clarification. Can you point to specific justifications and follow-ups?

How stable is the CURSOR score across different random splits for shuffled baselines? Since the score is computed via a relative RMSE ratio, do you observe high variance depending on the shuffling?

**Ethical Concerns:**

["NO or VERY MINOR ethics concerns only"]

**Final Justification:**

The paper provides an interesting contribution. The authors have addressed many of my and other reviewers' concerns. If the point of clarity are addressed in the final version of the paper, this can be accepted.

**Limitations:**

CURSOR enables mental target reconstruction using EEG, which raises important ethical concerns around cognitive privacy and unintended inference. The possibility of inferring internal mental content without explicit user intention should be acknowledged. A brief discussion of these risks and possible safeguards (e.g., user consent, secure model deployment, opt-in constraints) would strengthen the paper's ethical framing.

**Quality:**

2

**Strengths And Weaknesses:**

Strengths:

- Proposes the first self-calibrating BCI that operates in continuous latent spaces (e.g., 512D GAN embeddings) instead of discrete labels. Enables mental target recovery without calibration, labels, or pre-trained decoders.

- Introduces a relative RMSE-based score comparing aligned and shuffled EEG-stimulus datasets. This breaks the circular dependency between unknown distances and regressors. An interesting approach.

- Demonstrates high alignment with human perception, accurate ranking and successful optimisation to sub-perceptual distances.

- An EEG dataset with continuous latent-space stimuli and task-specific engagement.

Weaknesses:

- The introduction is not really clear and fails to explain the core problem and contributions of the paper. While it gestures at mental target recovery and self-calibration, it does not clearly explain how CURSOR works or what makes the problem novel. Terms like "continuous domains" and "mental targets" are used without grounding the reader in how EEG signals and image embeddings are being connected. Key elements, like the role of the latent space, the unsupervised regression formulation or the scoring mechanism, are absent or vague. As a result, the actual technical framing only becomes understandable much later in the method section, which weakens accessibility for readers who are not experts.

- The abbreviation CURSOR (Consistency-based Unsupervised Regression for Similarity-based Optimization and Ranking) feels contrived and only loosely tied to the actual method. While the method does involve unsupervised regression and ranking based on similarity scores, the notion of "consistency" is not a clearly defined or central mechanism in the algorithm. There is no consistency regularisation or self-distillation and there is only a relative RMSE comparison between aligned and shuffled EEG–stimulus pairs. Additionally, optimisation and ranking refer to downstream uses of the score, not core components of the algorithm itself.

- Line 159 claims that "This logarithmic sampling aligns with neuroscientific methodologies for studying perceptual thresholds and neural adaptation". This needs to be cited.

- Line 161 mentions “mitigations for these biases” but neither defines the biases nor explains the mitigations later. There is no clear follow-up. This statement is unsubstantiated and should be clarified or removed.

- The justification for the two-stage evaluation, ranking in full space and optimisation in reduced-dimensional space, feels weak and underexplained. The authors cite computational challenges as the reason, but provide no concrete analysis of resource constraints or runtimes. More importantly, reducing the dimensionality (from 512D and 203D to 10D and 20D) changes the problem structure and may artificially boost optimisation performance. This shift is not examined and the paper treats it as a benign workaround rather than a methodological compromise.

- The paper is difficult to read and hard to follow.

---

> ### Author Rebuttal · Authors · 2025-07-29
>
> We appreciate Reviewer Dn3x’s comprehensive and constructive commentary, which allowed us to deepen our analysis (“stability across different random splits”) and expand our ethics discussion.
>
> ## Q1: Can you justify the need for optimisation in reduced-dimensional spaces beyond computational cost? Does optimisation in 10D/20D reliably reflect performance in the original 512D/203D latent spaces?
>
> **Why report optimization in a low‑dimensional subspace?** While computational constraints motivated our choice (we had to run more than 8000 optimization experiments for our controls in Appendix Figure 6), it is considered standard to use dimensionality reduction as part of a black‑box optimization procedures [constantine2014,letham2020,harkonen2020,shen2021].
>
> **Why 10D/20D?** Our decision to use 10D/20D is fully supported by Appendix Figure 5. We ran the ranking task for different EEG dimensions (using PCA). At 20D, we observed the same performance. Then, maintaining EEG at 20D, we reduced the image space (via PCA), and 10D maintained similar performance. Performance was measured once images were converted back to their original 512D to ensure apples‑to‑apples comparison with ranking and human studies. In other words, we used the smallest dimensionality that preserves task performance (20‑D EEG, 10‑D faces) consistent with low‑effective‑dimension assumptions in common frameworks [constantine2014,eriksson2020,letham2020]. We will clarify this in the Appendix section C.
>
> **Is this a limitation?** We acknowledge this as a limitation in L328, and to our knowledge, it is not possible to guarantee the optimization performance directly on the full 512D search space based on our reduced-dimensional results. The limitation is not in using dimensionality reduction per se, as all our metrics are reported after faces are projected back to their original 512D (using PCA inverse mapping) to ensure apples‑to‑apples comparison with ranking and human studies. Instead, it is because we relied on external metrics to choose 10D/20D, which would not be available in a real scenario (i.e., correlation of scores vs ground-truth on the ranking task). Future research could study performance using internal dimensionality reduction metrics instead (e.g., reconstruction metrics).
>
> - [constantine2014] Active Subspace Methods in Theory and Practice: Applications to Kriging Surfaces
> - [harkonen2020] GANSpace: Discovering Interpretable GAN Controls
> - [shen2021] Closed-Form Factorization of Latent Semantics in GANs
> - [eriksson2020] Scalable Global Optimization via Local Bayesian Optimization
> - [letham2020] Re-examining linear embeddings for high-dimensional Bayesian optimization
>
> ## Q2: Line 159 (logarithmic sampling) and line 161 ("bias mitigation") lack citations or clarification. Can you point to specific justifications and follow-ups?
>
> **Logarithmic sampling.** This is standard practice in BCI experiments to acquire more data close to where information is maximal. In our case, the changes in brain activity are more sensitive to stimulus changes near the target [watson1983,wichmann2001,leopold2005,laming2010].
>
> - [watson1983] Quest: A Bayesian adaptive psychometric method
> - [wichmann2001] The psychometric function: I. Fitting, sampling, and goodness of fit
> - [laming2010] Fechner's law: where does the log transform come from?
> - [leopold2005] The dynamics of visual adaptation to faces
>
> **"Bias mitigation".** While this is standard and was required in our data collection to ensure data quality, this is a current limitation of our work that will need to be addressed before expanding to online closed-loop settings. We nonetheless stress‑tested for acquisition bias by:
>
> - (i) using uniformly sampled hypotheses for our ranking task evaluation, so they are decoupled from the log‑biased acquisition and
> - (ii) performing ablation studies, removing all near‑target items within set distances of the target. Appendix Figure 6 shows that performance starts to significantly degrade compared to a non-ablated baseline once d_removed>10 (N=5432), which indicates the possibility that a non-logarithmic sampling might be an approachable proposition during an online self-calibrating experiment.
>
> ## Q3: How stable is the CURSOR score across different random splits for shuffled baselines? Since the score is computed via a relative RMSE ratio, do you observe high variance depending on the shuffling?
>
> Thank you very much for this question, which allowed us to deepen our analysis.
>
> **Do you observe high variance depending on the shuffling?** Standard deviation of the scores in the 10-fold cross-validation is about 2.5% of the mean at N=9234 for most methods, except MLP at about 4%.
>
> **Additional signal-to-noise evaluation.** We were curious and studied further the difference between the range of scores and the average variance, as it would indicate how robust the rankings are. For this, we computed a "signal variation to measurement uncertainty ratio" that is:
>
> ```python
> np.std(np.mean(scores, axis=1)) / np.mean(np.std(scores, axis=1))  # [scores are (60, 10) dimensions for the ranking task]
> ```
>
> which is a signal-to-noise (SNR) type metric suitable for our ranking task.
> Interestingly, for S-LR and MLP, this SNR remains the same value around 0.4 for all N. LR and SVR start at the same value but linearly increase with N to significantly better values of 0.6 and 0.52, respectively, at N=9234.
>
> Analyzed with the correlation results in Figure 3 (left) and a plot of the mean+std scores, it explains well why MLP performance is underwhelming compared to LR and SVR. MLP scores correlate with ground truth distances, but the cross-validation variance on the scores is much higher than for LR and SVR. The SNR can capture this as a lower ratio, which in turn explains the considerable variability in d-top-rank performance in Figure 3 (right).
>
> We will add three Figures to the Appendix, along with the above analysis, to support this:
>
> - Evolution of STD as % of scores per method per N. This shows an exponential decay from 20% to 2.5% for most methods (4% for MLP).
> - Mean+std scores vs true distances aggregated for all runs---one plot for each method.
> - Signal variation to measurement uncertainty ratio (SNR) vs N for each method.
>
> ## A brief discussion of these risks and possible safeguards (e.g., user consent, secure model deployment, opt-in constraints) would strengthen the paper's ethical framing.
>
> Thank you for encouraging us to strengthen our ethical framing, which we do not take lightly.
>
> CURSOR links EEG to hypotheses about internal visual content; it raises the possibility of inferring mental states without explicit user intention. Physiological data can be sensitive and re‑identifiable, and consent mechanisms are often opaque in practice, which heightens the ethical salience of cognitive privacy for BCI research [solove2005taxonomy,ohm2010broken,kokolakis2017privacy,davis2024physiological].
>
> The core risk is that models or datasets are repurposed beyond the stated task to infer mental content without informed, ongoing consent. To address this, we will expand our privacy section in the Appendix to advocate for:
>
> - Acknowledging risk explicitly: We will state that inferring internal mental content without user intent is unacceptable and a central ethical concern. [boire2000cognitive,sententia2004neuroethical]
> - Requires opt‑in, revocable consent: Participation must be explicit, comprehensible, and withdrawable, with data erasure upon request. [berg2001informed,beauchamp2003methods,nissenbaum2004privacy]
> - Imposes purpose limitation and governance: Models and data may only be used for the approved task and context, with new consent required for secondary use and audits recorded. [european2016regulation,regulation2024ai,california2020cpra]
> - Implements data minimization and secure handling: Prefer on‑device or sandboxed processing; encrypt data in transit/at rest; set short retention windows; prohibit sharing of raw EEG. [malin2013biomedical,price2019privacy]
> - Increases transparency and accountability: Release model cards and datasheets documenting intended use, limits, failure modes, and oversight plans. [mitchell2019model,gebru2021datasheets]
> - Centers user agency: Provide a visible "record/mute" control, require explicit start/stop for any recording, and disable inference outside the study context by default. [davis2024physiological]
> Furthermore, our study is offline, task‑bound, and conducted with consenting volunteers; we do not perform real‑time or covert inference, and outputs are analyzed in aggregate, which we will explicitly detail as mitigations in the Appendix.
>
> [solove2005taxonomy] A Taxonomy of Privacy.
> [ohm2010broken] Broken Promises of Privacy: Responding to the Surprising Failure of Anonymization.
> [kokolakis2017privacy] Privacy Attitudes and Privacy Behavior: A Review.
> [davis2024physiological] Physiological Data and Privacy in the Wild.
> [boire2000cognitive] Cognitive Liberty and Mental Privacy.
> [sententia2004neuroethical] Neuroethical Concerns, BCIs, and Cognitive Autonomy.
> [berg2001informed] Informed Consent.
> [beauchamp2003methods] Methods and Principles in Biomedical Ethics.
> [nissenbaum2004privacy] Privacy as Contextual Integrity.
> [malin2013biomedical] Biomedical Data Privacy and Security.
> [price2019privacy] Privacy in the Age of Machine Learning.
> [mitchell2019model] Model Cards for Model Reporting.
> [gebru2021datasheets] Datasheets for Datasets.
> [european2016regulation] GDPR: General Data Protection Regulation.
> [california2020cpra] California Privacy Rights Act.
> [regulation2024ai] EU AI Act.

---

> > ### Comment · Reviewer_Dn3x · 2025-08-05
> >
> > Thanks for providing the rebuttal and addressing a lot of my concerns. I have recommended an accept.

---

> > > ### Author Response · Authors · 2025-08-06
> > >
> > > Thank you for taking the time to review our work. We are grateful for the constructive comments that have helped strengthen the clarity and presentation of the work.

---

### Official Review · Reviewer_c1xC · 2025-06-30

**Clarity:** 3
**Significance:** 3
**Originality:** 3
**Rating:** 5
**Confidence:** 2

**Summary:**

This work introduces a self-calibrating framework that recovers mental targets (e.g., imagined faces) from EEG and image data without using labeled data or pre-trained models. In particular, the framework generates a hypothetical mental target as an embedding of a face the user might be imagining and constructs a dataset that relates observed faces to their distances from this target. It then trains an estimator to predict these distances from EEG data and compares its performance to that of an estimator trained on a mismatched dataset. The resulting score reflects how well the hypothesis matches the true mental target, with the score expected to peak when the hypothesis aligns with the actual target the user had in mind. Through this mechanism, one can predict perceptual similarity, rank images relative to an unknown target, and generate target-like images. The authors also release a new EEG dataset, face stimuli, and codebase to support future research.

**Questions:**

1. Did you experiment with non-linear estimators? I have my doubts a linear estimator is the best choice for this problem.
2. For the dummy condition, in L239 the authors say “average of all distances in Gamma set”. Is this average computed in the training split or all the samples?
3. I miss results with at least one supervised method from prior work. Can you include some? I understand the premise here is to avoid labeled data but it would be good to show the gap between the presented approach and the current label-based methods in terms of accuracy.
4. In table 1, I would like to see top-k accuracy as well to have a better intuition of the performance. The distance metric is somewhat hard to use for comparison. Accuracy would also be needed if item 3 is included.

**Ethical Concerns:**

["NO or VERY MINOR ethics concerns only"]

**Final Justification:**

The authors have addressed my questions and added additional results that reinforce the initial conclusions. Furthermore, in my opinion, they also adequately addressed other reviewers' concerns.

**Limitations:**

Yes

**Paper Formatting Concerns:**

No concerns.

**Quality:**

3

**Strengths And Weaknesses:**

A major strength of this work is its introduction of a label-free method for mapping EEG to image data through self-calibration, overcoming the limitations of prior approaches that depend on labeled EEG-image pairs. Another key contribution is the release of a paired EEG and image dataset, which can support and encourage further research in this area.

Although I’m not deeply familiar with standard practices in BCI, I found it particularly clever that the authors operate in the latent space of images—this enables manipulation of ground-truth distances prior to the generation of observable data.

As acknowledged by the authors, a primary limitation of the approach lies in the search procedure: the framework restricts the hypothesis space to a predefined set that includes the true mental target. This makes the framework effective as a discriminator, but as a classifier, it requires manual curation and prior knowledge to ensure the target is within the search set. I also raise some concerns about the evaluation, which I elaborate on in the Questions section below.

---

> ### Author Rebuttal · Authors · 2025-07-30
>
> We are grateful to Reviewer c1xC for a thorough review and for praising the novelty and contribution of our work. Following their recommendations, we have significantly improved our manuscript by rerunning experiments that yield additional insights on CURSOR’s performance (top-k accuracy).
>
> ## Q1: Did you experiment with non-linear estimators? I have my doubts that a linear estimator is the best choice for this problem.
>
> Yes. In addition to linear regression (LR), we evaluated a multilayer perceptron (MLP) and support vector regression (SVR) with an RBF kernel under the same training/validation protocol (L224). Results for all estimators are summarized in Table 1, with a detailed breakdown in Appendix Figure 8. We also tested a non‑linear EEG representation using EEGNet embeddings as inputs; results are reported in Appendix Table 2. Across these settings, linear regression achieved the best performance.
>
> Please also see our comment to Reviewer VJPj ("Using CURSOR with more complex estimators") regarding the estimator's absolute performance. As CURSOR computes a relative score, the absolute performance of an estimator on the ground truth data might not always be the most significant factor. Understanding this relationship would constitute interesting future research, which Appendix Figure 8 started to explore.
>
> ## Q2: For the dummy condition, in L239 the authors say “average of all distances in Gamma set”. Is this average computed in the training split or all the samples?
>
> The training split. We keep everything the same (i.e., the cross-validation procedure is the same) and simply swap the estimator for the dummy estimator. The estimators thus only see the training set as for all other conditions. We further emphasize that aligned and shuffled RMSEs are computed on identical held‑out folds; thus, the dummy estimator will always give the same prediction for the aligned and shuffled sets, leading to a CURSOR score of 1. This is the most basic baseline possible to verify that no bias or information leaks exist in our pipeline.
>
> ## Q3: I miss results with at least one supervised method from prior work. Can you include some? I understand the premise here is to avoid labeled data but it would be good to show the gap between the presented approach and the current label-based methods in terms of accuracy.
>
> Thank you for sharing this concern. We kindly refer this Reviewer to our answer to Q3 from Reviewer m8FY.
>
> ## Q4: In table 1, I would like to see top-k accuracy as well to have a better intuition of the performance. The distance metric is somewhat hard to use for comparison. Accuracy would also be needed if item 3 is included.
>
> We re-ran experiments to compute top-k accuracies for the LR method and compare with the theoretical top-k for random retrieval (out of 60 candidates). Please see the table below:
>
> | Model                       | Top-1 Acc      | Top-3 Acc      | Top-5 Acc      | Top-10 Acc     |
> |----------------------------|-------------------|----------------|----------------|----------------|
> | LinearRegression           | 0.171 ± 0.029  | 0.353 ± 0.037  | 0.565 ± 0.038  | 0.800 ± 0.031  |
> | Theoretical Random Retrieval | 0.016 ± 0.01 | 0.049 ± 0.016  | 0.081 ± 0.021  | 0.155 ± 0.028  |
>
> **With respect to Q3 (“item 3”)**, our hypothesis set does not have an EEG attached to them, and we thus could not rank them based on a pre-trained decoder. A significant strength of CURSOR is that it can estimate the score of any hypothesis face $h$ based on a separate dataset of (eeg, z) pairs. A decoder-based scorer, trained on (eeg, d) pairs, would need access to one (eeg, h) pair to be able to score that specific $h$.
>
> **Why do we report d-at-top-rank and not top-k?** We chose to report the d-at-top rank in our initial submission (Figure 3 and Table 1) because we think it is more informative for our perceptual task. If the top-1 face is not the correct target but is visually similar to the target (i.e., its distance to the target is small, see H-ID experimental results Figure 2 - right), we would argue that reporting that distance is more informative than whether it is the correct answer. In addition, our hypothesis distance to the target was sampled uniformly (L185); thus, the rank, while a very good proxy, does not necessarily reflect perceptual similarity.
>
> We will add a table for min-d-at-top-k in the Appendix. This is the minimum distance to the target in the top-k retrieved hypothesis, informing us whether the top-k retrieved set contains an image perceptually identical to the target when combined with the H-ID results.
>
> The table below compares the LR method with the theoretical top-k for random retrieval (out of 60 candidates):
>
> | Model                       | Top-1 Min Dist  | Top-3 Min Dist  | Top-5 Min Dist  | Top-10 Min Dist |
> |----------------------------|-----------------|------------------|------------------|------------------|
> | LinearRegression                         | 2.898 ± 3.067   | 1.010 ± 1.312    | 0.524 ± 0.861    | 0.176 ± 0.450    |
> | Theoretical Random Retrieval | 23.064 ± 13.316 | 11.532 ± 8.933   | 7.688 ± 6.498    | 4.194 ± 3.828    |

---

> > ### Comment · Reviewer_c1xC · 2025-08-04
> > **Thank you**
> >
> > Thank you for addressing my questions, providing additional results, and clarifying my concerns. In light of the superior performance of this method across several metrics, as well as its flexibility due to its unsupervised paradigm, I believe this is a valuable contribution to the community. I have reviewed the other reviewers' comments and I am satisfied with the authors' rebuttal. For all these reasons, I am raising my score accordingly.

---

> > > ### Author Response · Authors · 2025-08-06
> > >
> > > Thank you for taking the time to review our work. We are grateful for the constructive comments that have helped strengthen the clarity and presentation of the work.

---

### Official Review · Reviewer_VJPj · 2025-07-01

**Clarity:** 3
**Significance:** 1
**Originality:** 3
**Rating:** 3
**Confidence:** 5

**Summary:**

This paper proposes a self-calibrating BCI to retrieve face images from raw EEG responses projected to a lower-dimensional space. Instead of training the system for classification using labelled data, which is customary, it exploits an estimator to predict distances to potential targets in the EEG domain, with the estimator trained using distances between face images in an image latent space. An error function is used as a measure of predictability to locate a target image, an image that a human participant has first looked at and then been asked to remember, while a batch of images is quickly shown to the participant, and EEG responses are recorded.

**Questions:**

What does it mean that the proposed algorithm is self-calibrating? In what sense are other algorithms calibrated? Calibration typically means that you determine a set of parameters using a set of known examples, and then keep these parameters fixed in further experiments. What would those parameters be here?

Why would it be a novelty to work in continuous domains, and what does this mean in this context? The only relevant domains are the image domain, the latent image domain, and the EEG domain. Which of these are typically not continuous?

How is the iterative search being done? Iterative over what?

**Ethical Concerns:**

["NO or VERY MINOR ethics concerns only"]

**Final Justification:**

Given the additional information provided in the rebuttal, which would have been nice to have seen in the actual paper, there is no reason to change the recommendation.

**Limitations:**

yes

**Paper Formatting Concerns:**

There seems to be no concerns related to the formatting.

**Quality:**

2

**Strengths And Weaknesses:**

This paper introduces the first self-calibrating BCI for continuous domains, which is shown in experiments to work well for mental target retrieval using face images as targets. A dataset of as many as 9234 image-EEG pairs collected from 29 subjects is also presented. A user study is also being conducted, from which a strong correlation between the proposed scores and perceived similarities could be observed.

It is mentioned seven times on the first two pages that the proposed framework is the first self-calibrating BCI for continuous domains. The question is whether this is indeed the case. The problem formulation and assumptions in this paper seem to be very similar to those in [1], which contains a system for iteratively searching for a target face image by gradually locating a subregion in the image latent space using sequences of EEG responses. This paper does not assume labeled training examples either, which is true for other earlier examples referred to in the same paper.

The proposed error function, with its associated score function, measures how well an estimator can be trained to predict distances to a given target from EEG, using a set of paired images and EEG responses. The assumption is that if the error is small, the target image is likely to be correct, but if the target is incorrect, the error ought to be larger. From the experimental results, this seems to be a valid assumption, at least for an estimator based on linear regression, but not at all for one based on a multi-layer perceptron. The problem with this argument is that it will never work with more competent estimators.

Modern neural networks can be trained to predict any function, even if the distances in the training data are completely random, as long as the amount of data is limited, which is true in this case. For such a network, you are thus likely to get almost no errors at all. That is probably why the linear regressor seems to be preferred in this case. It could be that the distance function becomes more linear when the target is correct, not that the error becomes smaller and the predictability higher.

It is claimed that the proposed method includes an iterative search for a target, but it is not described how this search is done. Section 4.3 is vague indeed and includes few details. There is nothing that excludes the possibility that the actual search is indeed brute force through either all examples or a reduced set of examples.

Unfortunately, there seem to be no experiments on any real face images. The retrieved images are almost exactly the same as the target one, if everything works well, but this is highly unrealistic in a real context, when you use anything beyond a limited set of generated images.

[1] Rajabi et al., “Mental Face Image Retrieval Based on a Closed-Loop Brain-Computer Interface”, HCII, 2023.

---

> ### Author Rebuttal · Authors · 2025-07-29
>
> We thank Reviewer VJPj, especially for the questions on the framing of self‑calibration, which prompted productive discussions among the authors. We will clarify misunderstandings, notably regarding Rajabi et al. [1], which includes a *supervised, per‑subject classifier‑training phase*.
>
> ## Differences with Rajabi et al [1]
>
> We disagree that [1] “does not assume labeled training examples either”. In [1] section 4.2 says “the classifier is trained for each subject separately using their EEG data collected while performing a target face detection task (Sect. 5.2)”, and, in section 5.2, it is clear that the process is a “two-phase experiment: first, we train the EEG classifier for each subject (the classifier task) and subsequently we evaluate the performance of our closed-loop system in a mental face image retrieval task (the closed-loop task)”.
>
> Reusing the same elements of language as in [1], our paper introduces a method that can start tackling the “closed-loop task” directly, without requiring the prior “classifier task”. Specifically, [1] explicitly includes a supervised, per‑subject classifier task trained on labeled EEG before the closed‑loop retrieval phase. Our method addresses the closed‑loop task directly and is label‑free/self‑calibrating. This is the core contribution of our paper and is, to the best of our knowledge, novel.
>
>
> ## Q1: What does it mean that the proposed algorithm is self-calibrating? In what sense are other algorithms calibrated? Calibration typically means that you determine a set of parameters using a set of known examples, and then keep these parameters fixed in further experiments. What would those parameters be here?
>
> **What is self-calibrating, and how is CURSOR self-calibrating?** We say CURSOR is self-calibrating because we do not need an explicit calibration phase to get labels.
>
> ---
>
> In a **calibrated** approach, typical stages are;
>
> Phase 1: Calibration:
>
> 1. collect EEG responses to images of faces, while the participant keeps a specific face in mind (the target face is **known to the experimenter**). Here we collect (eeg, z) pairs and know $z^*$.
> 2. compute labels based on the distance to the known target face ($d = ||z^*, z||_2$),
> 3. train a decoder $f_\theta(eeg) \mapsto d$ and freeze its parameters $\theta$,
>
> Phase 2: Downstream task:
>
> 4. use the frozen decoder for inference on downstream tasks
>
> ---
>
> With **self-calibration**;
>
> Phase 1: Self-calibration:
>
> 1. collect EEG responses to images of faces, while the participant keeps a specific face in mind (the target face is **unknown to the experimenter**). Here we collect (eeg, z) pairs, but **we do not know $z^*$**.  Then, recover the target face via CURSOR. Here we estimate $\hat{z}$ as our best guess for z*. This is the core contribution of the paper. Our results support that $\hat{z}$ is close to z* (Figure 3 - right and Figure 4 - left).
>
> 2. compute labels based on the distance to the estimated target face ($d = ||\hat{z}, z||_2$). Our result (Section 7: Recovering ground-truth labels) shows we can recover ground-truth labels with minimal error. **Our paper stops here, as once labels are recovered, it is technically trivial to follow with**:
> 3. train a decoder $f_\theta(eeg) \mapsto d$ and freeze its parameters $\theta$,
>
> Phase 2: Downstream task:
>
> - 4. use the frozen decoder for inference on downstream tasks
>
> ---
>
> **What parameters are being estimated?** With the above in mind, CURSOR itself does not have ”persistent” parameters that are being estimated and then frozen for later use. It would be interesting research to generalise a CURSOR estimator for a specific task/domain based on a large unlabelled dataset so that $f_{\theta}(\Gamma, h) \mapsto scores$ for any $\Gamma$ and $h$. $\theta$ would constitute “persistent” parameters for a generalized CURSOR estimator.
>
>
> **Manuscript changes.** We will explicitly refine the definition of *self‑calibrating* in the Introduction to address the above.
>
> ## Q2: Why would it be a novelty to work in continuous domains, and what does this mean in this context? The only relevant domains are the image domain, the latent image domain, and the EEG domain. Which of these are typically not continuous?
>
> It is novel for self-calibrating systems, not for BCI tasks. We discuss this in L29-L32, and it is detailed in the related work section.
>
> Prior self‑calibrating pipelines operated on finite, discrete stimulus sets (e.g., letters, colors, flashes) with binary feedback from EEG (match/non‑match), and framed calibration as a classification problem. By contrast, our method calibrates without labels over a continuous latent stimulus space (hypotheses $h \in \mathbb{R}^{512}$) using a continuous‑valued feedback as the latent‑space distance $\rho_h(z_i)=\lVert h-z_i\rVert_2$.
>
> **What “continuous” means here:**
> - **Stimuli:** We search a continuous latent stimulus space rather than a finite list of categories.
> - **Feedback information:** EEG encodes real‑valued distances $\rho_h(z_i)$, not binary labels.
>
> **Consequence for algorithm design (why this is non‑trivial).** Continuous stimulus space and continuous‑valued feedback require (i) regression instead of classification and (ii) hypothesis‑centric search over an effectively infinite set.
>
> **Positioning and claim.** To our knowledge, this is the first label‑free, self‑calibrating BCI that performs closed‑loop search over a continuous latent stimulus space using a continuous‑valued feedback signal. We will revise the Introduction to state the claim at this level of precision.
>
>
> ## Q3: How is the iterative search being done? Iterative over what?
>
> Iterative search is the framework we use to position our work.
>
> In the iterative search framework, scores are computed iteratively as new EEG-Image pairs are collected **while interacting with a user**. This is why we report most results with N (we iterate over N) as the x-axis, showing how performance changes as we iteratively acquire more data.
>
> Referring to the wording used in [1], we call “iterative search”, what [1] calls a “closed-loop task”, with the key difference that our evaluation is **offline** (iterative but non‑interactive) with EEG recorded during a **pre‑generated** sequence, and all iterative analyses are performed post‑hoc.  We also do not perform any **active sampling** in our EEG and computational experiments, with the following samples selected randomly from the remaining data. This differs from typical “closed-loop” systems.
>
> We will ensure the difference with “closed-loop” systems is clear in our related work section in the paragraph starting L74 and cite [1] as a supporting example.
>
>
> ## Using CURSOR with more complex estimators
>
> - “it will never work with more competent estimators.”
> - “Modern neural networks can be trained to predict any function, even if the distances in the training data are completely random, as long as the amount of data is limited”
>
> We kindly disagree with these statements. Potential overfitting issues are accounted for by (1) using cross-validation to compute scores, ensuring we do not train and test on the same data, and (2) using relative scores against a shuffled control (i.e., completely random training data), ensuring overfitting or poor-performing estimators are neutralized.
>
> **Cross-validation.** With proper cross‑validation, any improvement that generalizes to held‑out data typically reduces the aligned error more than the shuffled error, increasing $S(h)$ for hypotheses consistent with the true target while leaving inconsistent hypotheses near the null $S(h)\approx 1$. Hence, a stronger estimator does not break CURSOR.  Empirically, **we already include a stronger/nonlinear model** variant (e.g., EEGNet feature extractor + MLP head in Appendix Table 2) and observe that performance is not degraded.
>
> **Relative scores.** CURSOR never ranks hypotheses using raw predictions alone; it uses a **relative error improvement** computed on test splits via cross-validation. If a model performs similarly on aligned than on shuffled labels (due to poor performance or overfitting), then $S(h)\approx 1$ for all $h$, which could be detected and used to prevent an early decision.
>
> **Manuscript changes.** We will further emphasize that aligned and shuffled RMSEs are computed on identical held‑out folds to clear up any confusion about overfitting and leakage.
>
>
> ## No experiments with real faces
>
> **What we did (scope and limitation).** We would appreciate it if the Reviewer could clarify what they meant by “no experiments on any real face images”. Our EEG study used GAN‑generated, photorealistic face stimuli as proxies for photographs. We acknowledge that we did **not** collect EEG responses to photographs of real identities, and we will state this limitation in the revision, as we already discussed generator/latent‑space constraints in the paper.
>
> **Generate vs. retrieve (why near‑duplicates can appear).** Our method does not retrieve from a fixed gallery; it selects a latent hypothesis and then generates the image via the pretrained generator. When the selected hypothesis is very close to the target in latent space, the synthesized output can be perceptually indistinguishable from the target image (as per our H-ID experiment, Figure 2 - right), yet it is not an exact copy, confirmed by subtle differences, e.g., in background details.
>
> Images generated under Linear Regression (LR) remain perceptually close to the target, whereas baseline methods (S‑LR, Dummy) that participants in H-ID could identify as different from the target. This qualitative pattern is consistent with the quantitative H‑ID metrics in Figure 2. We will ensure Section 7 reports these observations more explicitly.

---

> > ### Comment · Reviewer_VJPj · 2025-08-08
> >
> > Thanks for the information provided in the rebuttal, in particular regarding the definition of self-calibration, iterative search and the importance of the relative score measure. The promised modifications to the paper are much appreciated. Unfortunately, the unconventional use of concepts like self-calibration and iterative search in the paper makes it rather hard to understand.
> >
> > It is true that modern neural networks cannot really model any function, incorrectly stated in the review, but they can be trained to interpolate any training data, as long as the capacity of the network is larger than the complexity of the data. However, the linear regressor, that showed to perform do well in this study, does not have such a high capacity. That is why you observe a difference in score between aligned and shuffled data.
> >
> > Experiments on real face image data would have been interesting to see, given that the space of generated images provided by a generator tends to be smaller than the space of real face images. Other methods tend to perform worse on real face images and it would not be surprising if this were the case for CURSOR as well.

---

> ### Author Response · Authors · 2025-08-06
>
> Dear Reviewer VJPj,
>
> Following the Program Chairs' reminder regarding the author-reviewer discussion period, we wanted to check in briefly. If you find the time, we would greatly appreciate your feedback on whether our responses have addressed your concerns, and we remain available to provide further clarification if helpful.
>
> \
> With best regards,
>
> Submission #6015 Authors

---

> ### Author Response · Authors · 2025-08-08
>
> Thank you for the positive feedback that helped us clarify our use of the terms self-calibration and iterative search.
>
> **High capacity estimators.** Regarding modern neural networks, we remain a bit confused about the exact issue. We think the reviewer refers to a possible overfitting issue, which is commonly exacerbated by high-capacity estimators (e.g., deep networks). However, *we kindly refer back to our rebuttal comment*: we never tested on the training set (we always compute estimates via 10-fold 90/10 cross-validation) which should eliminate such issues.
>
> In addition, *Appendix Table 2 in the paper* shows that a modern neural network embedding layer (EEGNet) followed by a fully connected network (BestMLP) performs at a level equivalent to a Linear Regressor when used in CURSOR. Specifically, the associated metrics for EEGNet-encoded EEGs are:
>
> | Regressor | d top rank | R      | Rank   |
> |-----------|------------|--------|--------|
> | LR        | 2.66 ± 3.02 | -0.76 ± 0.05 | 6.21 ± 6.03 |
> | S-LR      | 21.07 ± 13.73 | -0.01 ± 0.13 | 26.35 ± 16.93 |
> | BestMLP   | 3.91 ± 3.53 | -0.76 ± 0.04 | 6.14 ± 5.89 |
>
> We will update Section 7 in the paper for clarity on this point. *Does such data-driven evidence help clear this issue?*
>
> **Real faces.** We agree that experiments on real faces would constitute interesting future research. The core challenges would be: 1) a new EEG experiment to acquire a novel dataset of stimuli-responses pairs, and 2) the introduction of a new component to the algorithm, which allows a quantitative distance definition that correlates with faces' visual similarity.
>
> ---
>
> Thank you for taking the time to review and provide feedback on our work. We are grateful for the constructive comments that have helped strengthen the clarity and presentation of the work.

---

### Official Review · Reviewer_m8FY · 2025-07-04

**Clarity:** 2
**Significance:** 2
**Originality:** 3
**Rating:** 5
**Confidence:** 2

**Summary:**

The paper presents an approach to recover the image a participant holds in their mind given their EEG brain data in response to multiple images that differ more or less from the target image. This approach does not require labelled data and relies on previous research showing that distance between visual concepts can be predicted from the EEG. The authors collected a new EEG dataset on 31 participants, in which images of faces generated with a GAN (previously trained for this purpose) are presented in an RSVP oddball paradigm. The proposed algorithm, CURSOR, is then used to predict the distance (in latent space) between each shown image and a series of hypothesis images, given EEG responses. Results suggest that the version of the algorithm based on linear regressors enables retrieving or reconstructing the target image, or a perceptually indistinguishable variant, as opposed to shuffled/dummy baselines.

**Questions:**

1. I don’t understand how CURSOR can be evaluated on all subjects and trials at once. From what I understand, the score depends on how well the distance between a hypothesis image and a presented image can be modelled from EEG. If this distance is well modelled (i.e., high RMSE ratio), then the hypothesis image is a good candidate. However, since the target image differs from trial to trial, this process must be run for each trial individually. In this case, what does it mean to scale the dataset size beyond the number of related images available in a trial (e.g. in Figure 3)? I believe I must be missing something obvious here.
2. What is the trajectory the related images are sampled from (line 157), i.e., is another point in latent space randomly sampled and used to provide the direction?
3. This is purely a suggestion to help clarify how CURSOR differs from a supervised approach (conceptually, for the reader, but also in terms of performance) where labels are available. Would it be possible to integrate one such approach as a baseline? For instance, by training a predictor in leave-one-subject-out (or leave-one-trial-out) fashion that learns on ($e_i$, $z^*$) pairs, and comparing its ranking/predictions to the model.

Small comments

* I assume that each channel is averaged across time within each time window, however it is not specified in the text, e.g. line 167.
* Appendix B says there are 16 target images, but Section 5 (l156) says there are 17.

**Ethical Concerns:**

["NO or VERY MINOR ethics concerns only"]

**Final Justification:**

The authors have addressed my concerns and clarified the functioning of their proposed approach during the discussion period. Their experimental results overall seem to support the claims, and their proposed label-free approach is novel. With this in mind I raise my score to an accept, and suggest to the authors that they include the content of their clarification answers in the manuscript.

**Limitations:**

Yes.

**Quality:**

3

**Strengths And Weaknesses:**

Strengths
* Quality: The submission appears technically sound, with claims supported by experimental results. For instance, the alignment between the CURSOR score and human judgment of distances is first validated, as well as the distance threshold for perceptually similar images (Figure 2). The comparison between the proposed linear regression-based model and controls (a version with a dummy regressor or a version trained on a shuffled dataset) supports the feasibility and effectiveness of the approach. Finally, the examples shown in Figures 4 and 10 overall support the quantitative results.
* Significance: I am not familiar enough with the specific problem of mental visual target recovery to evaluate the impact this approach may have on the community. However, given the new unsupervised paradigm, and unique accompanying dataset, this submission is likely to help design further studies on this topic.
* Originality: the submission’s approach to the recovery of mental visual targets appears to be novel, as is the data that has been collected as part of the study. Relevant previous work is cited, and the difference with existing paradigms is described (see Q3 for a suggestion to further clarify this).

Weaknesses

* Clarity: I found the manuscript overall difficult to navigate, though this may be due to me being less familiar with the topic. A key point that I think should be clarified is how CURSOR can be evaluated on all subjects and trials at once, i.e. what a N=9234 actually means (see Q1). Also, it was not obvious from the main text and appendix that the total number of examples (9234) was obtained by only keeping the 28.5% of oddball images from the 1120 images shown in a trial.

---

> ### Author Rebuttal · Authors · 2025-07-30
>
> We appreciate Reviewer m8FY’s rigorous critique which directly contributed to improving the clarity of our revision.
>
> ## Q1: How can CURSOR be evaluated on all subjects and trials at once? + dataset scaling clarification
>
> The neurophysiological data were indeed acquired in controlled conditions during which we collected triplets of $(z_i,e_i, z_i^\*)$, where $z^\*$ is the target we asked the user to keep in mind during each experiment. This is explained in Section 5 L168: "Our final dataset consists of 9234 stimuli-response pairs, with each entry composed of a $z^\*_i$ vector in R512, a stimuli $z_i$ vector in R512, and an EEG response vector $e_i$ in R203". Seventeen different targets were used during our neurophysiological data acquisition, and the Reviewer is correct that, as is, we could not use the entire dataset for our experiments.
>
> **How we “scaled the dataset size beyond the number of related images available”**
> We needed $(z_i,e_i)$ pairs all associated with the same target to have enough data. Given previous work confirmed that $e_i$ encodes $\lVert  z^\* - z_i \rVert_2$  [delatorreortiz2023], we could generate arbitrary new datasets $\psi_{N}$ containing N=9234 $(z_i,e_i)$ pairs where the EEG responses remained associated with the original target-stimuli distance (i.e., preserving the original statistical properties).
>
> [delatorreortiz2023] The P3 indexes the distance between perceived and target image
>
> This is explained in Section 6.1: “Not all stimuli-response pairs in the dataset were collected with the same target face $z^\*$. To enable our study, we generated equivalent datasets, each with a single target face $z^\*$, by generating new stimuli in the latent Z-space that preserved the original statistical properties. We used gradient descent, starting from a randomly sampled 512-dimensional starting point, to match the Euclidean distance from the target in the original dataset to the new target. We created 10 dataset variants for each of the 17 target faces utilized in our neurophysiological experiments, yielding 170 stimuli-response datasets that preserved the original statistical properties”.
>
> **Implication:** We are mixing EEG data from different participants and use them as if coming from a unique participant, which might make the self-calibration harder to solve. A significantly larger dataset attached to a single participant would need to be collected and tested to evaluate such claims.
>
> **Changes to the paper.** We understand our text could be confusing and will update the opening sentence of Section 6.1 to make it more concrete.
>
>
> ## Q2: What is the trajectory the related images are sampled from?
>
> Trajectories move “diagonally” in the 512D-dimensional latent space, i.e. equally far in every coordinate direction. We simply add a small positive value to each component of the source image at monotonically increasing $d$ values with logarithmic spacing (denser near 0). We will clarify at L157 directly.
>
> For context, this is standard practice in face research, with the rationale that moving along one latent direction yields small, smooth, and monotonic changes in facial appearance, reducing uncontrolled feature leaps [valentine1991,rotshtein2005,kahn2010,gratton2013]. Adhering to such established best practices decreases risk in the data acquisition experiment, promoting data quality.
>
> - [valentine1991] A unified account of the effects of distinctiveness, inversion, and race in face recognition
> - [kahn2010] Temporally distinct neural coding of perceptual similarity and prototype bias
> - [rotshtein2005] Morphing Marilyn into Maggie dissociates physical and identity face representations in the brain
> - [gratton2013] Attention Selectively Modifies the Representation of Individual Faces in the Human Brain
>
> ## Q3: Would it be possible to integrate one supervised approach as a baseline? For instance, by training a predictor in leave-one-subject-out (or leave-one-trial-out) fashion that learns on (e_i, z*) pairs, and comparing its ranking/predictions to the model.
>
> Thank you for the suggestion. We discussed this internally prior to paper submission and decided against this direction as we were concerned that an emphasis on absolute performance might overshadow the contribution of our self-calibrating (label-free) approach.
>
> To avoid this issue while still allowing a fair comparison, we decided to report the reconstruction error of the ground truth target (Section 7 “Recovering ground-truth labels” and Appendix F6) as a proxy to calibration performance for any possible downstream tasks and methods. The rationale is that if self-calibration allows for accurately reconstructing the ground-truth labels, then any downstream supervised methods can be applied with their own specific requirements, constraints, etc.
>
> The reconstruction of labels can be done using the target identified via self-calibration as $\hat{d} = ||\hat{z}, z||_2$. Our label‑free CURSOR pipeline reconstructs these distances with RMSE = 0.18 $\pm$ 0.08 (Appendix F6). For context, a supervised decoder on the same EEG task and featurization (29 channels x 7 time windows, 203 features) has been reported in [delatorreortiz2024] with RMSE = 0.17 $\pm$ 0.09 on their aggregated evaluation. We will add this side‑by‑side comparison in the Appendix and note that cross‑validation meta‑parameters differ slightly across papers.
>
> [delatorreortiz2024] Perceptual Visual Similarity from EEG: Prediction and Image Generation
>
>
> ## Small Comments (SC)
>
> **SC 1: I assume that each channel is averaged across time within each time window, however it is not specified in the text, e.g. line 167.**
>
> You are correct, within each of the 7 windows, we average the samples across time for each channel (after baseline correction), producing one scalar per (channel, window); stacking across 29 channels and 7 windows gives the 203‑D vector used by the estimators. We will make this explicit in Section 5.
>
> **SC 2: Appendix B says there are 16 target images, but Section 5 (l156) says there are 17.**
>
> It is 17 indeed. Thank you, we will fix the typo.

---

> > ### Comment · Reviewer_m8FY · 2025-08-06
> >
> > Thank you to the authors for the additional explanations and clarifications. The responses have addressed my concerns and I will raise my score accordingly.

---

> > > ### Author Response · Authors · 2025-08-06
> > >
> > > Thank you for taking the time to review our work. We are grateful for the constructive comments that have helped strengthen the clarity and presentation of the work.

---

### Note · Authors · 2025-08-15

Once again, we thank the reviewers c1xC, Dn3x, m8FY, and VJPj for their positive comments, constructive discussion, and detailed suggestions. We have improved the clarity and presentation of our manuscript and will reflect these updates in the camera-ready version.

---

### Decision · Program_Chairs · 2025-09-17

**Decision:**

Accept (poster)

**Comment:**

This submission proposes a method for recovering a participant's mental target from their EEG and image data. The main contribution of this work is that they do not use label data or pre-trained models. The authors collected a new EEG dataset from 31 participants while they viewed synthetically generated images of faces. The proposed method is then used to predict the distances between the shown image and a set of images, as well as generate a new image that is close to the mental target.

After the author-reviewer discussion period, most of the reviewers recommended acceptance of this work. One reviewer (Reviewer VJPj) expressed concern about the proposed "error function" and the underlying motivation for its use. There were also concerns about the clarity of the submission and the use of synthetic faces; however, these were less critical. During the AC-reviewer discussion period, the other reviewers wrote that they do not see this as a major issue, rather one that needs further clarification. Therefore, provided the authors further clarify the logic motivating their choice of error function, I recommend acceptance.